# Heterogeneous plasticity of amygdala interneurons in associative learning and extinction

Natalia Favila [1,6], Jessica Capece Marsico [1,6], Catarina M. Pacheco [1,2], Selin Kenet [1], Benjamin Escribano [1], Yael Bitterman[3,4], Jan Gründemann [1,2], Andreas Lüthi [3,5] & Sabine Krabbe [1,3] ✉

Neural circuits undergo experience-dependent plasticity to form long-lasting memories, but how inhibitory interneurons contribute to this process remains poorly understood. Using miniature microscope calcium imaging, we monitored the activity of large amygdala interneuron populations in freely moving mice during fear learning and extinction. Here we show that interneurons exhibit complex and heterogeneous plasticity at both single-cell and ensemble levels across memory acquisition, expression, and extinction. Analysis of molecular interneuron subpopulations revealed that disinhibitory vasoactive intestinal peptide (VIP)-expressing cells are predominantly activated by salient external stimuli, whereas the activity of projection neuron targeting somatostatin (SST) interneurons additionally aligns with internal behavioural states. Although responses within each interneuron subtype are non-uniform, molecular identity biases their functional role, producing weighted circuit outputs that can flexibly regulate excitatory projection neuron activity and plasticity. These findings demonstrate that inhibitory interneurons actively shape the encoding and stability of emotional memories, underscoring their importance in adaptive learning.

Associative learning enables an organism to link environmental stimuli with their behavioural relevance. This is particularly important under conditions of immediate threat, such as in fear learning. One of the key brain regions regulating the acquisition, expression and extinction of conditioned fear behaviour is the amygdala, a highly conserved temporal lobe structure consisting of distinct subnuclei. Traditionally, excitatory projection neurons (PNs) of the cortex-like basolateral amygdala (BLA) were regarded as the main site of plasticity during memory formation[1]. Yet, learning processes are strongly influenced by dynamic shifts in the balance between excitatory and inhibitory components within neuronal circuits. For example, fear learning reduces extracellular GABA levels[2] and modifies the expression of the GABA-synthesising enzyme GAD67, GABA receptors and their clustering

protein gephyrin in the BLA[3–5]. Furthermore, BLA inhibitory circuits show diverse forms of synaptic plasticity ex vivo[6–9], which might be involved in both fear and extinction memory[10]. However, how interneuron activity changes during memory formation in vivo remains largely unexplored.

BLA interneurons are highly diverse and form intricate microcircuits. Distinct interneuron subpopulations can be distinguished based on marker gene expression, morphology, pre- and postsynaptic connectivity or functional properties, all of which are considered to be highly correlated with each other[11–13]. Even though they only represent about 15–20% of the neuronal population in the BLA[14], interneurons control PN activity in a spatially and temporally precise manner due to their cell type- and cellular compartment-specific postsynaptic

[1]German Center for Neurodegenerative Diseases (DZNE), Bonn, Germany. [2]Medical Faculty, University of Bonn, Bonn, Germany. [3]Friedrich Miescher Institute for Biomedical Research, Basel, Switzerland. [4]The Hebrew University of Jerusalem, Jerusalem, Israel. [5]University of Basel, Basel, Switzerland. [6]These authors contributed equally: Natalia Favila, Jessica Capece Marsico. ✉e-mail: sabine.krabbe@dzne.de

targeting. In consequence, selective changes in the activity patterns associated with memory formation of distinct inhibitory subpopulations can have diverse effects on PNs and ultimately amygdala output.

Recent studies started to delineate the contributions of interneuron subpopulations to fear learning and extinction. For example, somatostatin (SST) positive interneurons preferentially target the distal dendrites of BLA PNs and are thus ideally positioned to regulate synaptic inputs from thalamic and cortical sources[15]. Suppression of their activity during conditioning has been shown to enhance learning[15,16]. Furthermore, SSTs play a role in context-dependent fear suppression[17,18]. In contrast, interneurons expressing vasoactive intestinal peptide (VIP) contact other interneuron subpopulations, such as SST and parvalbumin (PV) positive cells[19,20]. When activated by aversive events during associative fear conditioning, VIP Interneurons have disinhibitory effects on BLA PNs and can thereby enable excitatory plasticity[19]. In addition, in vitro studies have demonstrated cellular plasticity of distinct interneuron subpopulations upon fear and extinction learning[21–23], highlighting that inhibitory cells undergo experience-dependent plastic changes themselves. However, only a few studies recorded the activity of individual neurons during fear learning and extinction. Recent work demonstrated learning-associated plasticity of distinct molecular interneuron subpopulations at the population level using fibre-photometry recodings[24]. Yet, even within the well-defined paradigms of associative fear learning and extinction, individual interneuron subpopulations display high heterogeneity[15,18,19,25]. However, due to the sparsity of BLA interneurons and the resulting low cell numbers per animal in in vivo recordings, a systematic classification of response patterns at the single-cell level across interneuron subpopulations is still lacking. Furthermore, a characterisation of interneuron plasticity would require to reliably record from the same cells over the course of several days, which is difficult to achieve in deep brain regions such as the amygdala.

To address this gap, we employed deep-brain calcium imaging with implanted lenses and miniature microscopes in freely behaving mice during an associative learning paradigm. By targeting all BLA inhibitory cells, we were able to follow large populations of interneurons at single cell resolution across days and provide a classification of their response types during fear learning, memory expression and extinction across molecular subtypes. These data reveal that BLA interneurons show complex plastic responses at both the ensemble and single neuron level, with distinct cells being selectively activated or inhibited upon fear or extinction training. Notably, interneurons suppressed by fear cues or activated during extinction are further modulated by internal behavioural state, suggesting a role in transitions out of fear or in safety signalling. Using this functional classification, we further demonstrate that VIP and SST interneurons show biased activity during fear learning and extinction, with VIP cells responding predominantly to salient external cues and SST neuron activity additionally reflecting internal states. Although both subtypes exhibit diverse responses at the single-cell level, their molecular identity shapes a weighted output signal for microcircuit computations that can modulate BLA long-range output and plasticity in fear and extinction.

## Results

### Deep-brain imaging of amygdala interneurons
To record the activity of identified BLA interneurons in freely behaving mice at single-cell resolution, we employed a gradient refractive-index (GRIN) lens-based imaging approach in combination with miniaturised microscopes[19,26] (Fig. 1a). Cre-dependent, virally mediated expression of GCaMP6f[27] in the BLA of *GAD2-Cre* mice[28] allowed for interneuron-specific Ca[2+] imaging. Immunohistochemical analysis revealed that all major interneuron subpopulations (SST+, PV+, VIP+ and CCK+ interneurons) were targeted with this approach (Fig. 1b and S1), matching fractions previously reported for the general inhibitory population in

the BLA[11]. Mice with head-mounted miniature microscopes underwent a 4-day auditory fear conditioning and extinction paradigm (Fig. 1c, "Methods"). For habituation, mice explored context A and were presented with two different auditory cues (6 kHz and 12 kHz pure tones) used as CS+ and CS− in the fear conditioning session on the next day. For conditioning, the CS+ was paired with an aversive US in the form of a mild foot shock for five times in a different context B, while intermingled CS− presentations were used as control tones. For test and extinction days in context A, the CS− was presented four times, followed by 12 CS+ stimuli. The exact lens placements in BLA subnuclei were confirmed post hoc ex vivo and revealed that most implant sites were in the basal amygdala ($N = 6$ mice) while a minority was at the border between lateral and basal amygdala ($N = 3$; Fig. S2a, c). On average, we recorded $58 \pm 6$ BLA interneurons per mouse ($N = 9$ mice; Fig. S2b) stably within and across four days (Fig. 1d, "Methods"). BLA interneurons showed diverse spontaneous activity patterns, as well as cell-specific responses to auditory and aversive stimuli (Fig. 1e, f).

### Overlapping encoding of sensory stimuli in amygdala interneurons during fear learning
We initially focussed our analysis on the fear conditioning day. We could observe a diversity of cellular responses to the predictive CS+ cue, the aversive US and the neutral CS− control tone across neurons and animals (Fig. 2a–f). On average, CS+ and CS− led to a mild activation of BLA interneurons, while the aversive US induced a strong response (Fig. 2b). However, both inhibition and activation could be seen in individual cells for the different stimuli (Figs. 2a, f; 3a and S3a), with significantly more CS+ excited than inhibited cells ($n = 519$ cells; CS+ inhibited 22%, activated 36%) but similar proportions for CS− and US modulated cells (CS− inhibited 27%, activated 33%; US inhibited 36%, activated 40%). A majority of BLA interneurons responded to the auditory cues and the foot shock (Fig. 2c, d), with significantly higher fractions to the aversive stimulus ($76 \pm 2\%$) compared to the CS+ ($58 \pm 3\%$) and CS− tones ($60 \pm 2\%$; $N = 9$ mice). Overlapping sensory coding was found in subpopulations of interneurons that responded to combinations of the CS +, CS− or the aversive US, yet this was not enriched above chance level in the overall population (Fig. 2d, e), suggesting that auditory and aversive information remain largely segregated during fear conditioning instead of resulting in immediate non-linear integration. Furthermore, neurons responding selectively to individual stimuli were spatially intermingled with multisensory interneurons in the BLA, rather than locally clustered (Fig. 2g, h). No obvious differences were observed between animals with basal amygdala implant sites compared to mice with more dorsal lens placements at the border between lateral and basal amygdala (Figs. S2b and 2c, e). Together, these data show that interneurons across BLA subregions are strongly modulated by auditory and aversive stimuli during conditioning. Furthermore, the coincidence of these two signals at the single-cell level makes them ideal candidates for cellular plasticity during associative learning.

### Plasticity of interneuron responses during associative learning
Previous studies showed adaptive CS and US responses during learning for selected BLA interneuron populations[15,19,24,25]. However, a systematic classification of plasticity response types is still lacking, as previous results have been obtained using fibre photometry or single-cell approaches with low cell numbers. Therefore, we next investigated how CS and US responses of individual BLA interneurons change during fear conditioning. On average, interneurons displayed decreasing US amplitudes over the course of the five CS+/US pairings ($n = 519$; Fig. 3b), as previously reported for VIP and PV BLA interneurons[19,24]. However, this gradual decline was not statistically significant for the population average (Fig. 3e, f). Next, we used a K-means clustering approach to classify cellular responses of significantly modulated interneurons across the five US presentations

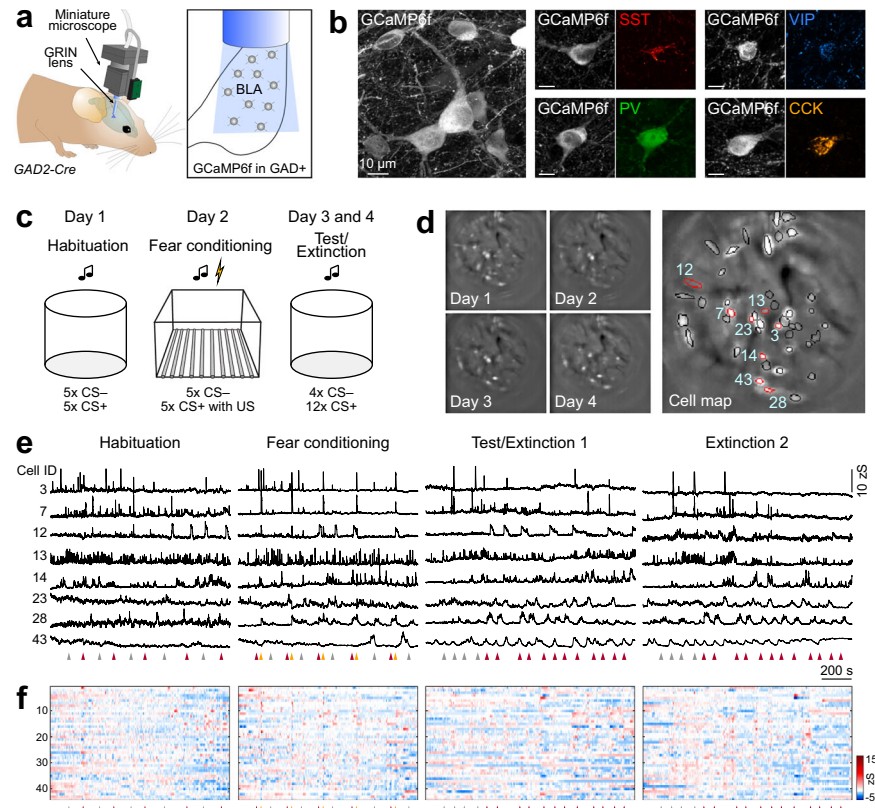

**Fig. 1 | Imaging interneuron activity in the basolateral amygdala of freely moving mice. a** Schematic of the approach used for deep-brain calcium imaging of BLA interneurons in freely behaving *GAD2-Cre* mice. Recordings were obtained using a miniature microscope after virus injection for Cre-dependent GCaMP expression and implantation of a gradient-index (GRIN) lens into the basolateral amygdala. **b** Confocal images of Cre-dependent GCaMP6f expression in BLA interneurons (representative of $N = 3$ mice, see Supplementary Fig. 1). Expression of all main interneuron markers was detected in GCaMP+ neurons (SST somatostatin, PV parvalbumin, VIP vasoactive intestinal peptide, CCK cholecystokinin). **c** Scheme of the 4-day discriminative auditory fear conditioning paradigm, consisting of habituation, fear conditioning, test and extinction sessions. **d** Individual motion corrected fields of view (maximum intensity projection) of one example animal across the 4-day paradigm (left) and the resulting cell map across all days (right). Circles indicate selected individual components. **e** Representative example traces of selected interneurons across the four-day paradigm (highlighted with red outlines in (**d**)). Arrowheads in (**e**) and (**f**) indicate starting points of CS+ (red), CS− (grey) and US (yellow). **f** Activity map of all identified interneurons from the example mouse in (**d**) across the entire paradigm ($n = 44$ cells).

($n = 393$; Fig. 3c, d, g). This revealed distinct types of activated and inhibited patterns, including cells with stable US signalling, but also interneurons that down- or up-regulate their US response with repeated pairings, as well as post-US activated ('US off') cells. Across animals, the 'Activated down' cluster was most prominent (29%), compared to smaller fractions particularly of the 'Activated up' (9%) and 'US off' patterns (7%; Fig. 3h). This analysis illustrates that the population average during the aversive stimulus (Figs. 2b and 3b) is dominated by the activated cells, although a comparable fraction of cells displayed inhibition of their intrinsic activity. Averaging these cluster representations across animals ($N = 9$) instead of using the recorded population ($n = 519$) yielded similar results (Fig. S4a, b), demonstrating that individual clusters were not dominated by single mice with high cell numbers.

We used a similar analysis approach to characterise CS responses of BLA interneurons, for which we could observe both activated and inhibited neurons across all five presentations of the CS+ and CS− ($n = 519$; Fig. S3a). On the population level, responses to both stimuli occurred during the initial presentations (Fig. 4a, d). However, at the end of the fear learning session, only the CS+ induced a clear activation, with a significantly stronger response at the fifth presentation compared to the subsequent CS− (Fig. 4a, d, g), indicating stronger upregulation of interneuron activity during the predictive cue compared to tones that are not paired with an aversive outcome. We next used a clustering approach (see "Methods") to define response types in individual interneurons that were significantly modulated by the CS+ ($n = 297$ cells) and

CS− ($n = 312$; Fig. 4b, c, e, f). This demonstrated that both predictive CS+ and control CS− tones induced cellular plasticity during the conditioning paradigm. Neurons that were only activated or inhibited by the first of these tones were equally represented for both conditions (Fig. 4h). Comparable fractions of BLA interneurons were initially not modulated by the respective CS but became activated ('Up' cluster) or inhibited during learning ('Down inhibited'). However, only for the CS+ a 'Stable activated' pattern emerged (24%). This was significantly different from the CS− that was not associated with stably activated cells. Instead, a fraction of interneurons selectively downregulated their activity after the second presentation ('Down activated', 9%). None of these clusters were dominant when comparing their proportions across animals (Figs. S3b, c and S4c−e). Together, these results suggest that the increased CS+ activity that develops at the population level at the end of the conditioning session (Fig. 4a, g) is not mediated by an upregulation of interneuron activity across trials, but achieved by maintaining a stable activation pattern during the session. In contrast, interneurons produce a balanced excitation and inhibition to the CS− during progressive learning, leading to no noticeable CS response at the population level at the end of training (Fig. 4d).

To investigate whether CS+ plasticity types depend on US activation, we further compared the US and CS+ response patterns of individual neurons during the fear conditioning paradigm (Fig. 4i). These data demonstrate that CS+ activity of single neurons evolves independently of their US activation, as adaptive CS+ responses were observed in interneurons activated and inhibited by the US, but also in

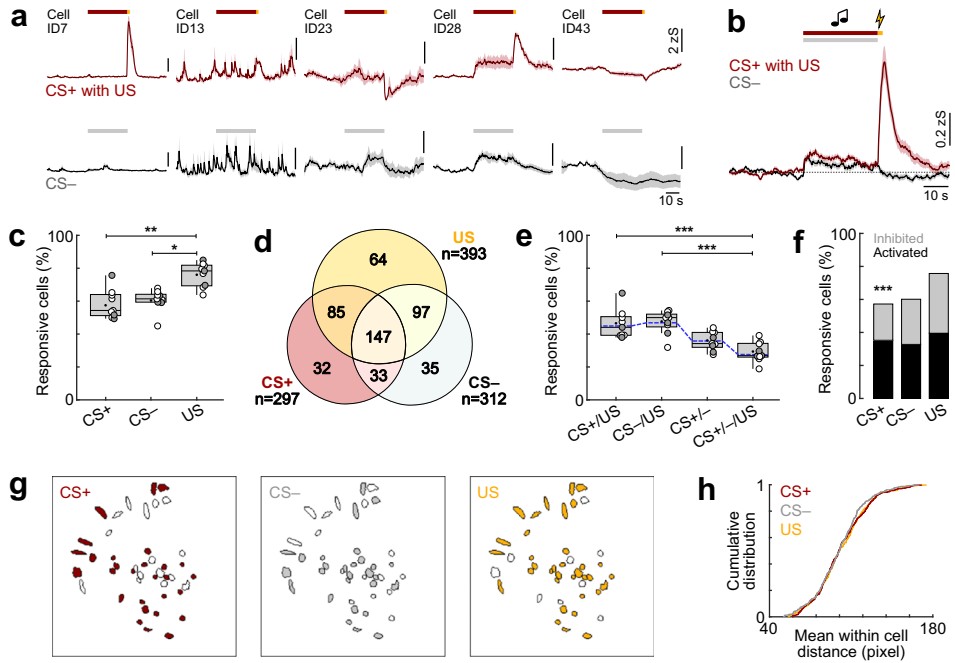

**Fig. 2 | Encoding of sensory stimuli in amygdala interneurons during fear learning. a** Representative example traces illustrating diverse activity patterns of BLA interneurons, averaged across five pairings of CS+ with US (top) or five presentations of the CS− (bottom). Red/grey lines indicate CS+/CS− duration, yellow line US. Cell IDs correspond to the recording shown in Fig. 1. **b** CS and US responses from all recorded BLA interneurons across five trials averaged across all mice (*N* = 9). **c** Fraction of interneurons responsive to the CS+, CS− and US across distinct animals (*N* = 9). Friedman test (χ2 = 11.53), *p* = 0.0016, followed by Dunn's multiple comparisons (CS+ vs. US, *p* = 0.0065; CS− vs. US, *p* = 0.0286). **d** Overlap of CS+, CS− and US responsiveness in individual interneurons. **e** Proportion of overlap in CS+, CS− and US activity in interneurons (*N* = 9). Blue line indicates chance overlap level. Friedman test (χ2 = 24.33), *p* < 0.0001, followed by Dunn's multiple comparisons (CS+/US vs. CS+/CS−/US, *p* = 0.0004; CS−/US vs. CS+/CS−/US,

*p* = 0.0002). **f** Percentage of BLA interneurons with significantly increased or decreased calcium responses during distinct stimulus presentations (*n* = 519; Chi-Square test with Bonferroni correction: CS+ inhibited vs. activated, *p* < 0.0001). **g** Example spatial maps of CS+, CS− and US coding neurons (same animal as in Fig. 1). **h** Cumulative probability distribution of the pairwise distance of CS+, CS− and US modulated neurons, showing no spatial clustering. Average traces (**a**, **b**) are mean with s.e.m.; Tukey box-and-whisker plots (**c**, **e**) show median, 25th and 75th percentiles, min to max whiskers with the exception of outliers, dots indicate mean. Circles (**c**, **e**) represent individual animals (open circles, imaging sites in basal amygdala, *N* = 6; filled circles, border of lateral/basal amygdala, *N* = 3; see also Supplementary Fig. 2). *\*p* < 0.05, *\*\*p* < 0.01, *\*\*\*p* < 0.001. See Supplementary Table 1 for statistics details.

cells without any detectable US response at the level of the soma. In summary, recording the activity of large BLA interneuron populations at the single-cell level during auditory fear learning revealed distinct plastic response types, both for the aversive US teaching signal but also for tone cues. The significant differences between CS+ and CS− illustrate the differential plasticity of BLA interneurons depending on the predictive value of a stimulus.

### Amygdala interneurons signal high and low fear states across conditioning and extinction

We next assessed how BLA interneuron CS responses change across days during fear expression and extinction. To this end, we first compared neuronal responses to the CS+ and CS− at the population level before conditioning (habituation), after conditioning (24 h after learning) and after extinction in the same behavioural context. For both CS+ and CS−, we could observe strong tone activation on the population level in the habituation session (*n* = 519 cells; Figs. 5a, d and S5), which overall decreased across the session. At the single-cell level, responsive neurons could be classified into activated and inhibited cells, with comparable proportions of these clusters between CS+ and CS− before conditioning (Fig. S5a–e). Across mice, we could not detect a difference in neurons activated or inhibited by the CS+ vs. CS− in habituation (Fig. S5f, g). Since 6 kHz and 12 kHz tones were counterbalanced as CS+ and CS− in the experimental cohort, we further analysed whether these frequencies induced differential activity in BLA interneurons (Fig. S5h–l). We could observe similar clusters of activated and inhibited neurons with no bias for either frequency.

Across days, no clear response of BLA interneurons could be detected after conditioning or after extinction for either CS on the population level (Fig. 5a, d). Yet, individual neurons showed significant activation or inhibition (Figs. S6a, b and 5g). Overall, proportions of neurons significantly activated or inhibited by the auditory cues were comparable between CS+ and CS− on all behavioural days (Fig. 5g). Clustering CS responses of significantly modulated cells across all days (CS+, *n* = 365; CS−, *n* = 357) revealed the emergence of distinct response types in individual interneurons (Fig. 5b, c, e, f). For both tones, these were associated with neutral tone presentations (activated or inhibited only during habituation), high fear states (only after conditioning) and low fear states (only after extinction). A subset of interneurons was further stably inhibited or activated across days, although the stably activated cluster could only be observed for CS− control tones (19%), suggesting that dynamic changes in CS coding across days are associated with aversive outcomes. While the predictive CS+ induced significantly stronger activation at the single-cell level after conditioning (CS+ 18%, CS− 8%) and after extinction (CS+ 17%, CS− 9%), for the CS−, a larger fraction of interneurons was in found in the 'Activated before conditioning' cluster (CS+ 18%, CS− 27%; Fig. 5h), an effect that could not simply be attributed to increased neuronal activation by the CS− in naïve mice before learning (Fig. S5f, g). For the CS−, this 'Activated before conditioning' cluster was further significantly enriched across all animals (*N* = 9 mice; Figs. S4f–h and S6f), whereas CS+ response types were evenly represented (Fig. S6e). Further, a comparison between responses during conditioning with fear and extinction coding revealed that this

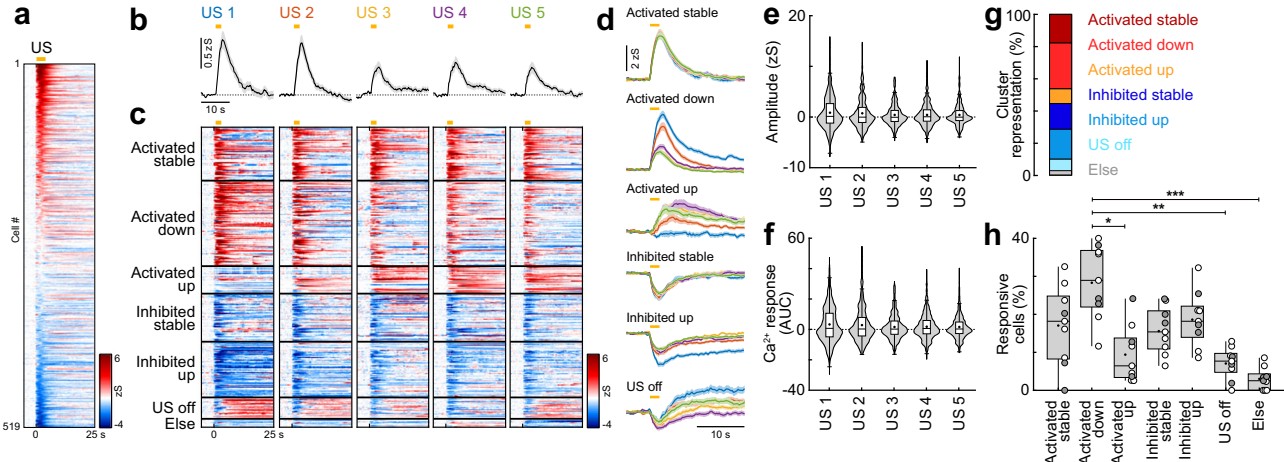

**Fig. 3 | Interneuron responses to aversive stimuli are highly diverse and plastic.** **a** Heatmap of basolateral amygdala interneuron responses to the aversive US (averaged across all five presentations) sorted by response amplitude (n = 519 cells from N = 9 mice). Yellow line indicates US duration. **b** Average traces of amygdala interneurons during the five aversive US presentations (n = 519). **c** Heatmap of single cell US responses clustered into groups depending on their US response pattern across the five presentations (n = 393 responsive cells; 'Activated stable', n = 70; 'Activated down', n = 112; 'Activated up', n = 36; 'Inhibited stable', n = 63; 'Inhibited up', n = 73; 'US off', n = 28; 'Else', n = 11). **d** Average traces of neuronal clusters in (**c**). **e** Average amplitude and **f** average area under the curve (AUC) during the five US presentations (n = 519). **g** Proportion of cells in US clusters.

**h** Fraction of interneurons according to US cluster membership across animals (N = 9). Friedman test ($\chi 2 = 30.41$), $p < 0.0001$, followed by Dunn's multiple comparisons ('Activated down' vs. 'Activated up', $p = 0.0327$; 'Activated down' vs. 'US off', $p = 0.0023$; 'Activated down' vs. 'Else', $p < 0.0001$). Average traces (**b**, **d**) are mean with s.e.m.; violin plots (**e**, **f**) show distribution of all data points; Tukey box-and-whisker plots (**e**, **f**, **h**) median, 25th and 75th percentiles, min to max whiskers with exception of outliers, dots indicate mean. Circles in (**h**) represent individual animals (open circles, imaging sites in basal amygdala, N = 6; filled circles, border of lateral/basal amygdala, N = 3). *$p < 0.05$, **$p < 0.01$, ***$p < 0.001$. See Supplementary Table 1 for statistics details.

plasticity of individual BLA interneurons is independent of their US activation during conditioning, indicating that a CS-US coincidence, at least on the level of somatic Ca²⁺ read-outs, is not necessary for memory-associated cellular plasticity across days (Fig. 5i).

Previous tracing and functional studies suggest that excitatory PNs mediating fear and extinction memory are preferentially located in the anterior vs. posterior parts of the BLA, respectively, and are defined by pre- and postsynaptic connectivity, in particular in relation to prefrontal cortex subdivisions[29–32]. To test for such an anatomical specificity of interneurons within the BLA, we mapped the location of CS+ response types onto horizontal atlas planes (Figs. S2c and S6c), which revealed no preferential location of fear or extinction interneurons. Together, this data indicates that individual interneurons across anterior and posterior BLA subregions develop prominent responses to predictive cues associated with fear learning and extinction, which could not be detected by monitoring a population average (Figs. 5a, d and S6d).

To reveal how neuronal responses to the conditioned CS+ change from a high fear state at the beginning of the test/extinction 1 session to a low fear state at the end of the extinction session 2, we plotted the cue-evoked activity of the CS+ fear and extinction clusters across these two sessions (Fig. 5j). Responses of both 'Fear' and 'Fear inhibited' interneurons gradually weakened during the first extinction session resembling intrasession extinction. Furthermore, these neurons exhibited weaker responses 24 h later at the beginning of the second session, which is reflected in spontaneous recovery at the behavioural level and followed by fast extinction. In contrast, extinction interneuron responses, both activated and inhibited, developed more rapidly at the end of the second extinction session. Complementary, we used K-means clustering of all neuronal responses (n = 519) across the two extinction sessions. This revealed similar functional clusters as found when focussing on neurons significantly modulated across habituation, test and extinction, namely fear activated, fear inhibited and extinction activated cells (Fig. S6g, h). In addition, a separate group of neurons was found to be activated predominantly in extinction session 1, while extinction inhibited cells were grouped in a cluster

inhibited across all CS+ presentations in extinction session 2. This suggests that the gradual extinction of fear memories is associated with additional dynamic changes in BLA interneuron population activity developing over the course of repeated CS+ presentations without the reinforcing US.

To examine the relationship between BLA interneuron responses and behavioural outcomes, we correlated the number of interneurons in each cluster with the behavioural performance (Fig. S7a–d). Mice with a higher number of CS+ 'Fear inhibited' interneurons showed significantly better discrimination between CS+ and CS−. Conversely, mice with a greater number of 'Extinction activated' interneurons tended to demonstrate poorer discrimination performance. This suggests that interneurons with suppressed activity after conditioning contribute directly to accurate threat discrimination by facilitating transient disinhibition during fear expression. On the other hand, the prevalence of extinction-activated interneurons may reflect individual differences in fear generalisation regulation, either as a pre-existing inhibitory bias or as a compensatory mechanism in animals with less precise learning.

Finally, we aimed to explore whether BLA interneurons responsive to conditioned auditory cues exhibit distinct activity reflecting internal fear states. To this end, we analysed their calcium dynamics during spontaneous immobility bouts in both extinction sessions in the absence of external stimuli, i.e. auditory cues. On average, interneurons displayed a decrease in activity upon immobility start, and an increase with movement onset after freezing stopped (n = 519; Fig. 6a). However, individual neurons showed heterogeneous responses, with a smaller proportion of cells being activated upon immobility start and inhibited with movement onset. We hypothesised that specific CS+ clusters activated by predictive cues in states of high or low fear, i.e. test and extinction timepoints, might correspondingly signal internal fear states during spontaneous freezing epochs. Therefore, we mapped the activity of fear and extinction interneuron clusters during immobility bouts in the intertrial intervals. While only minor changes in activity could be observed for CS+ 'Fear activated' and 'Extinction inhibited' interneurons, in particular 'Fear inhibited' and to a smaller

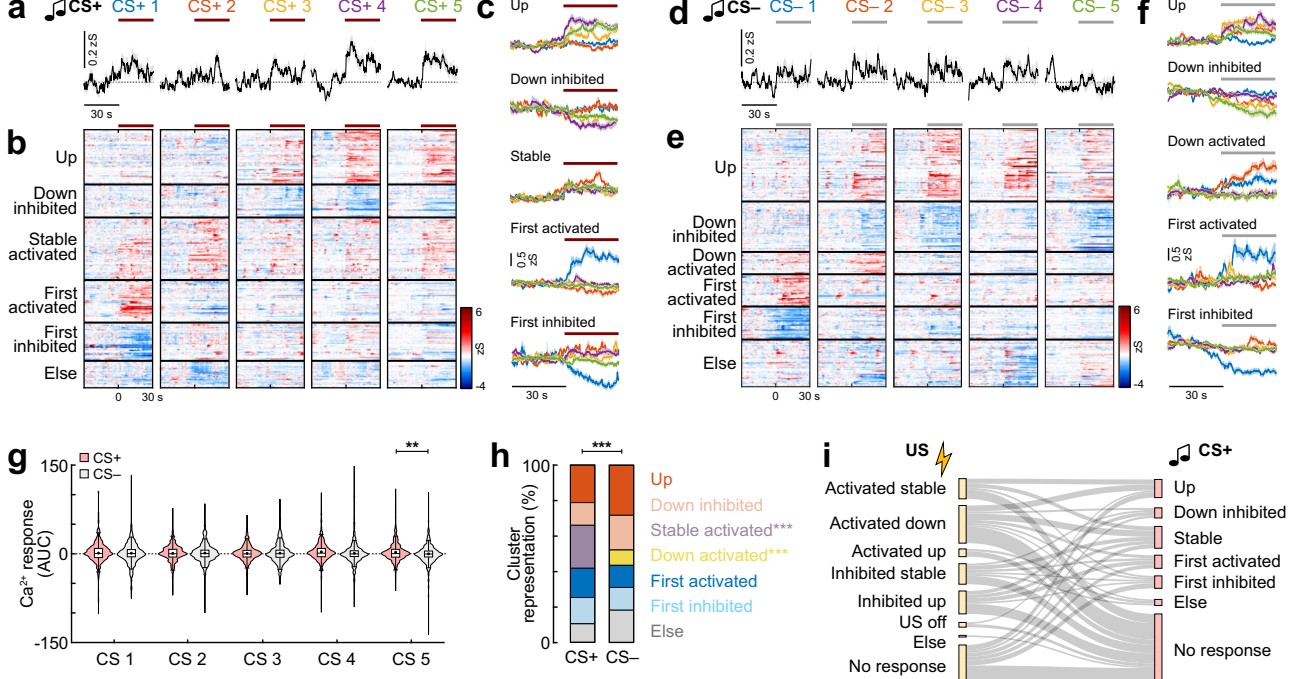

**Fig. 4 | Interneuron responses to auditory cues depend on the predictive value of the stimulus. a** Average traces of BLA interneuron activity during five presentations of the predictive CS+ during conditioning (n = 519 cells from N = 9 mice). Line indicates CS duration. **b** Heatmap of single cell CS+ responses clustered into groups depending on their CS+ response pattern across the five trials (n = 297 responsive cells; 'Up', n = 63; 'Down inhibited', n = 38; 'Stable activated', n = 72; 'First activated', n = 49; 'First inhibited', n = 44; 'Else', n = 31). **c** Average traces of neuronal clusters in (**b**). **d** Average traces of BLA interneurons during five presentations of the CS− control tone during conditioning (n = 519 cells). **e** Heatmap of single cell CS− responses clustered into groups depending on their response pattern across the five trials (n = 312 responsive cells; 'Up', n = 88; 'Down inhibited', n = 61; 'Down activated', n = 27; 'First activated', n = 39; 'First inhibited', n = 40;

'Else', n = 57). **f** Average traces of neuronal clusters in (**e**). **g** Area under the curve (AUC) for CS+ and CS− presentations during conditioning. Paired Wilcoxon test with Bonferroni correction; CS 5, CS+ vs. CS−, p = 0.0021, n = 519. **h** Proportion of cells in CS+ and CS− clusters (CS+, n = 297; CS−, n = 312). Chi-Square test CS+ vs. CS − (χ2(6) = 117.19), p < 0.0001; post hoc Chi-Square test with Bonferroni correction; 'Down activated', p < 0.0001; 'Stable', p < 0.0001. **i** Sankey plot illustrating the relationship of activity during the aversive US (Fig. 3) with CS+ plasticity patterns. Flows of less than 5 cells not shown. Average traces across panels are mean with s.e.m.; violin plots (**g**) show distribution of all data points; Tukey box-and-whisker plots median, 25th and 75th percentiles, min to max whiskers with exception of outliers, dots indicate mean. **p < 0.01, ***p < 0.001. See Supplementary Table 1 for statistics details.

degree 'Extinction activated' cells showed decreased activity with freezing start and an increase with movement onset (Fig. 6b, c). Importantly, these activity patterns were stable across days, as individual neurons showed consistent activity profiles during spontaneous immobility bouts throughout habituation as well as in extinction sessions 1 and 2 (Figs. 6d, e and S7e–h). These results indicate that BLA interneurons suppressed by auditory cues in fear, or activated by the CS+ during extinction, are also dynamically modulated by the animal's behavioural state. Their inhibition during spontaneous freezing and reactivation upon freezing offset suggests a role in encoding transitions out of fear or safety-related processes. In contrast, CS+ 'Fear activated' and 'Extinction inhibited' neurons appear less sensitive to internal state transitions, consistent with a role in representing cue-specific associative information rather than generalised fear levels.

### Inhibitory population activity changes day-to-day without losing overall stimulus representation

To determine if stimulus identity could be decoded from interneuron activity patterns, we trained multiclass decoders using binary linear support vector machines (SVM) on each training day[33,34]. To account for variations in cell population sizes between animals, we selected 37 cells randomly from each animal and report the average decoder accuracy of 100 independent iterations. Decoders accurately distinguished between CS+, CS−, and baseline using interneuron activity in all sessions, with a mean accuracy of 94 ± 3% (Fig. 7a). This accuracy was not due to better decoding of one class over another, as all three

classes showed high precision, recall, and F1 scores (see "Methods" for details), indicating a balanced performance on each day (Fig. S8d). Decoders trained solely on CS responsive interneurons (i.e. significantly responsive to CS+ or CS−, see Figs. 4 and 5) maintained similar accuracy (92 ± 4%). However, using non-CS responsive interneurons reduced the accuracy to 76 ± 11%, which was nevertheless higher than chance-level accuracy obtained from decoders trained on randomly shuffled labels (47 ± 0.2%). To eliminate the possibility that discrepancies in non-CS responsive interneuron decoder precision were attributable to a reduced number of available cells, we trained control decoders on a random subset of all interneurons, ensuring an equivalent number of cells to those used in the non-CS decoders (Fig. S8a).

We next assessed changes in CS+ and CS− encoding across days. To this end, we trained two-way SVMs to decode CS+ vs. baseline, and CS− vs. baseline on each day. These models were then used to decode stimuli identity on subsequent days. Although stimuli identity could be decoded within each day (mean intraday accuracy: CS+, 95 ± 4%; CS−, 95 ± 3%), the same interneurons could not decode stimuli identity on another day. Decoding accuracy dropped close to chance level to an average of 60 ± 3% for CS+ and 62 ± 3% for CS− when using a model trained on a different day (Fig. 7b). This decrease was mainly due to poor decoding of CS+ and CS−, as accuracy, precision, and F1 score were higher for baseline than for CS+ or CS− (Fig. S8e, f). This indicates that individual interneurons change dynamically from day to day, but information remains encoded at the population level on individual days.

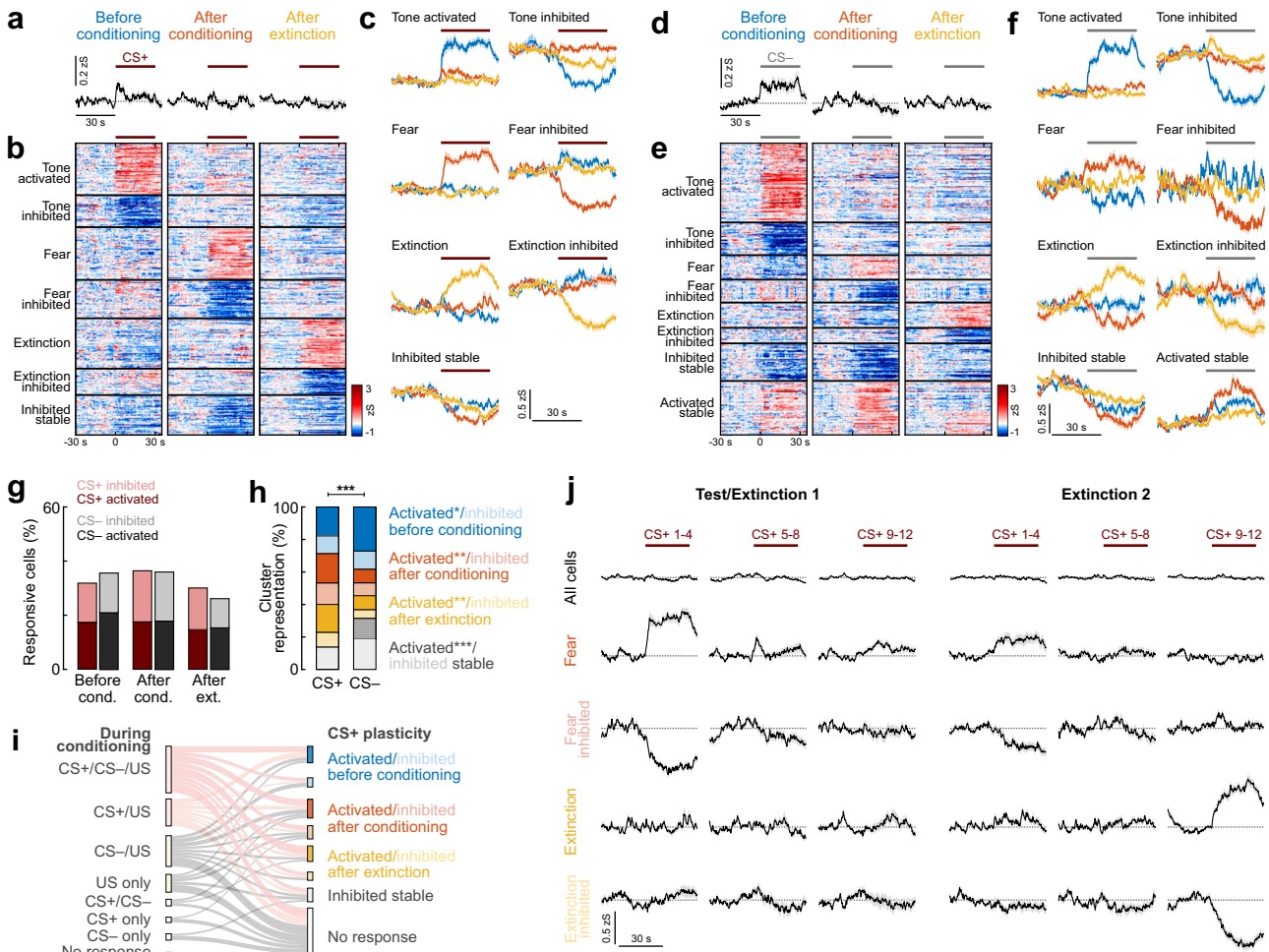

**Fig. 5 | Learning-related plasticity favours predictive cues in a heterogeneous interneuron population. a** Average traces of interneuron activity during the CS+ (*n* = 519 cells from *N* = 9 mice). Line indicates CS duration. **b** Heatmap of single cell CS+ responses clustered into groups (*n* = 365 responsive neurons; 'Tone activated'/ activated before conditioning, *n* = 65; 'Tone inhibited'/inhibited before conditioning, *n* = 40; 'Fear'/activated after conditioning, *n* = 66; 'Fear inhibited'/inhibited after conditioning, *n* = 48; 'Extinction'/activated after extinction, *n* = 63; 'Extinction inhibited'/inhibited after extinction, *n* = 33; 'Inhibited stable', *n* = 50). **c** Average traces of CS+ clusters in (**b**). **d** Average traces during the CS− (*n* = 519 cells). **e** Heatmap of clustered CS− responses (*n* = 357; 'Tone activated'/activated before conditioning, *n* = 97; 'Tone inhibited'/inhibited before conditioning, *n* = 40; 'Fear'/activated after conditioning, *n* = 30; 'Fear inhibited'/inhibited after conditioning, *n* = 28; 'Extinction'/activated after extinction, *n* = 31; 'Extinction

inhibited'/inhibited after extinction, *n* = 19; 'Inhibited stable', *n* = 46; 'Activated stable', *n* = 66). **f** Average traces of CS− clusters in (**e**). **g** Proportions of responsive neurons (*n* = 519). **h** Proportion of cells in CS+ and CS− clusters (CS+, *n* = 365; CS−, *n* = 357). Chi-Square test CS+ vs. CS− ($\chi 2(7) = 105.84$), *p* < 0.0001; post hoc Chi-Square test with Bonferroni correction; 'Activated before conditioning', *p* = 0.0275; 'Activated after conditioning', *p* = 0.0016; 'Activated after extinction', *p* = 0.0074; 'Activated stable', *p* < 0.0001. **i** Relationship of fear conditioning activity with across-day CS+ plasticity. Flows of less than 5 cells not shown. **j** Development of responses across extinction. All cells, *n* = 519; Fear, *n* = 66; Fear inhibited, *n* = 48; Extinction, *n* = 63; Extinction inhibited, *n* = 33. Average traces across panels are mean with s.e.m. *p* < 0.05, **p* < 0.01, ***p* < 0.001. See Supplementary Table 1 for statistics details.

To investigate interneuron selectivity for CS+ or CS−, we obtained the corresponding absolute decoding weights from the two-way decoders for each interneuron and calculated the correlation between them. If interneurons selectively encoded CS+ or CS−, one would expect a negative correlation between their weights, where a higher decoding weight for CS+ would corresponds to a lower decoding weight for CS− (see ref. 34). However, the correlation between CS+ and CS− decoding weights was close to zero after the conditioning session (Fig. 7c, d), suggesting that interneurons were not tuned to a stimulus and instead display broad tuning, consistent with our single cell analysis results (Fig. 2). Controls calculating the correlation between CS+ decoding weights showed high correlation (Fig. S8b, c).

Moreover, we evaluated how fear conditioning altered the differentiability of CS+ and CS− from the US as learning progressed. For population vector distance (PVD) analysis, we calculated the Euclidean distance between the evoked population vector responses to CS+/CS−

and the US for the multidimensional space of *n* interneurons in each individual mouse[35]. To probe whether the population evoked responses were getting closer or farther away as fear conditioning progressed, we normalised the PVD change to the distance between CS and US during the first pairing. We found that by the end of the session (pairings 4 and 5), the distance between CS+ and US, as well as CS− and US, had significantly decreased, averaging a reduction of 29% ± 14% and 27% ± 16%, respectively (Fig. 7e). This indicates that both stimuli's representations became closer to the US over the course of training. Distinct activation patterns of interneurons might contribute differently to this distance decrease between CS+ and US. Therefore, we re-ran the analysis while removing interneurons from different clusters based on their US and CS+ activity patterns during fear conditioning and calculated the difference between early (pairings 1–2) and late (pairings 4–5) fear conditioning. Removing interneurons of the CS+ clusters had no significant effect on the distance (Fig. 7f). Interestingly,

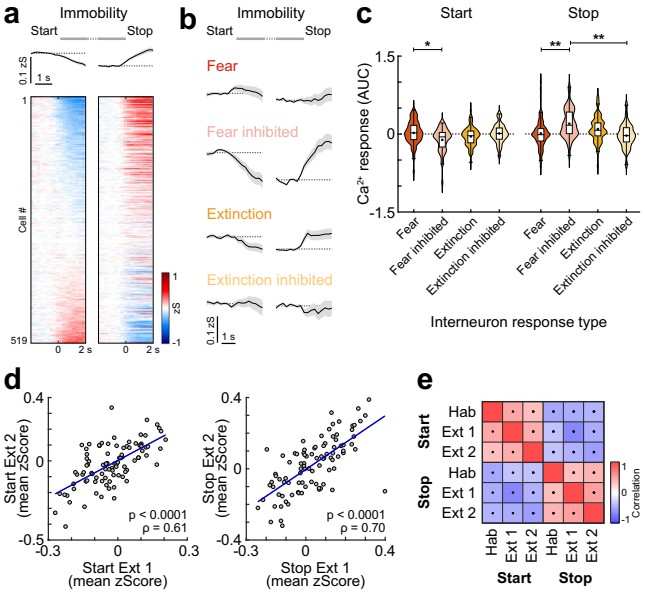

**Fig. 6 | Encoding of fear states in amygdala interneurons. a** Average traces and heatmap of basolateral amygdala interneuron activity during immobility bouts, averaged across all events in Test/Extinction 1 and Extinction 2 sessions, showing neuronal responses aligned to immobility start and movement onset (=immobility stop). Neurons are sorted by response amplitude upon immobility start ($n = 519$ cells from $N = 9$ mice). **b** Average traces sub-selected for across-day CS+ plasticity clusters during immobility (see Fig. 5; 'Fear'/activated after conditioning, $n = 66$; 'Fear inhibited'/inhibited after conditioning, $n = 48$; 'Extinction'/activated after extinction, $n = 63$; 'Extinction inhibited'/inhibited after extinction, $n = 33$). **c** Corresponding area under the curve (AUC) for immobility start and stop (2 s duration). Start: Kruskal Wallis test ($H = 10.83$), $p = 0.0127$ with post hoc comparison (Fear vs. Fear inhibited, $p = 0.0173$). Stop: Kruskal Wallis test ($H = 19.86$), $p = 0.0002$ with post hoc comparison (Fear vs. Fear inhibited, $p = 0.0015$; Fear inhibited vs. Extinction inhibited, $p = 0.0015$). **d** Example correlation of individual neuronal responses to immobility start (left) and stop (right) between Test/Extinction session 1 (Ext 1) and Extinction 2 (Ext 2), lines indicate linear regression fit (shown for visualisation); correlation coefficient $\rho$ reflects Spearman's rank-order correlation ($n = 93$ cells from mouse ID 710407). **e** Average correlation matrix of single cell responses to immobility start and stop across different stages of conditioning, dots indicate a significant difference of the correlation from 0 across $N = 9$ animals (Hab habituation, Ext extinction). Average traces (**a**, **b**) are mean with s.e.m.; violin plots (**c**) show distribution of all data points; Tukey box-and-whisker plots median, 25th and 75th percentiles, min to max whiskers with the exception of outliers, dots indicate mean. *$p < 0.05$, **$p < 0.01$. See Supplementary Table 1 for statistics details.

removal of interneurons of the 'Activated stable' US cluster decreased the distance between CS+ and US even more ('Activated stable', 32% vs. 'All cells', 26%; Fig. 7g), while removing interneurons with the 'Activated down' pattern increased the distance, albeit not significantly (17%). Thus, for the most part, the highly-defined interneuron response clusters for CS+ or US were not critical for the change in PVD between the CS+ and US during conditioning.

We further examined PVD changes across learning and extinction and found that the encoding for both CS+ and CS− with respect to the US remained overall stable, with average changes across days of $0.8 \pm 4.6\%$ and $1.2 \pm 3.9\%$, respectively (Fig. 7h). The CS− showed statistically significant fluctuations, moving closer to the US during fear learning, farther away after conditioning, and returning to baseline after extinction. However, these changes were minimal (before conditioning, $-3.5 \pm 4.2\%$; after conditioning, $1.5 \pm 2.6\%$; after extinction, $-1.6 \pm 3.0\%$) and exhibited high variance in individual animals. Thus, unlike BLA PN ensembles that display a lasting decreased in PVD and

thus an increase in the similarity of CS+ and US representations after conditioning[33], amygdala interneurons showed comparably stable representations of both CS+ and CS− across fear learning and extinction. Overall, our results indicate that BLA interneurons undergo heterogeneous plastic changes in single cell response patterns, yet representations of conditioned stimuli are stably encoded at the population level across fear learning and extinction.

## Molecular interneuron subpopulations contribute differently to the encoding of fear states

Finally, we aimed to identify how distinct molecularly defined interneuron subpopulations would contribute to the activity patterns we detected with the unbiased *GAD2-Cre* imaging approach. We chose to specifically characterise response dynamics in SST and VIP BLA interneurons, given their previously proposed opposing roles during fear learning[15,19,24] within the canonical VIP → SST → PN disinhibitory circuit motif[36,37] mediating dendritic disinhibition necessary for excitatory plasticity[15,38,39]. To this end, we performed experiments in *SST-Cre* and *VIP-Cre* mice and recorded these interneuron subpopulations across the learning paradigm (Figs. S9 and S10a–d). On average, we could reliably follow $29 \pm 5$ SST interneurons per animal ($N = 4$ mice; Fig. S9b) across the 4 days, and $25 \pm 3$ VIP cells per mouse ($N = 6$).

We first re-analysed the previously published dataset from the fear conditioning day[19] with a novel focus on response plasticity during learning. When comparing activity at the population level, stronger US activation was seen in VIP interneurons compared to SST cells (SST, $n = 114$ cells; VIP, $n = 101$ cells; Fig. 8a–c). The average US response was stable in SST interneurons across the five pairings with the CS+, while it decreased in VIP cells, as previously reported[19]. Yet, overall VIP activation was still significantly stronger compared to SST interneurons at the end of the session (Fig. 8b, c). At the single-cell level, a higher fraction of VIP interneurons was significantly activated by the US, and a larger proportion of SST cells significantly inhibited (Figs. 8g and S10e–h). To identify US response types in the two interneuron populations, we repeated the clustering of activity patterns based on the categories previously established with the unbiased imaging approach (Figs. 8h and S11a, b). Significantly more VIP interneurons were found in the 'Activated stable' group, which represented almost half of the VIP responses (SST 16%, VIP 48%). In contrast, the cluster 'Activated up', which gradually develops US responses during fear learning, was only detectable in SST interneurons (6%). While SST interneurons also showed higher fractions of US inhibited cells ('Inhibited stable': SST 26%, VIP 10%; 'Inhibited up': SST 17%, VIP 8%), the different cluster representation was not found to be statistically significant in our dataset. Overall, these results show that differences in aversive US coding between VIP and SST interneurons are mainly driven by stable activation and inhibition in these populations, respectively.

Next, we addressed how CS coding in BLA interneuron subpopulations changes during associative learning. We observed significant differences to the predictive CS+ between the SST and VIP subpopulations. At the population level, we found a stronger CS+ activation in VIP interneurons (Fig. 8d–f). This was reflected in stronger CS+ activation in individual VIP compared to SST cells, which were predominantly inhibited (Fig. 8g). Although SST cells appeared to be more activated across all five CS− presentations (Fig. S11e, f), differences in CS− activity could not be detected on the population level (Fig. S11g), nor in the fractions of significantly modulated cells (Fig. 8g). We further used the clustering approach to assign the previously determined classification of CS responses (Fig. 4) to the molecular interneuron subpopulations (Fig. S11c, d, h, i). For the CS+, we detected a significantly different cluster distribution between SST and VIP interneurons (Fig. 8i), with a higher fraction of stably activated neurons in VIP (SST 19%, VIP 49%), but a smaller fraction of cells inhibited by the first CS+ (SST 19%, VIP 3%). In contrast, no significant differences

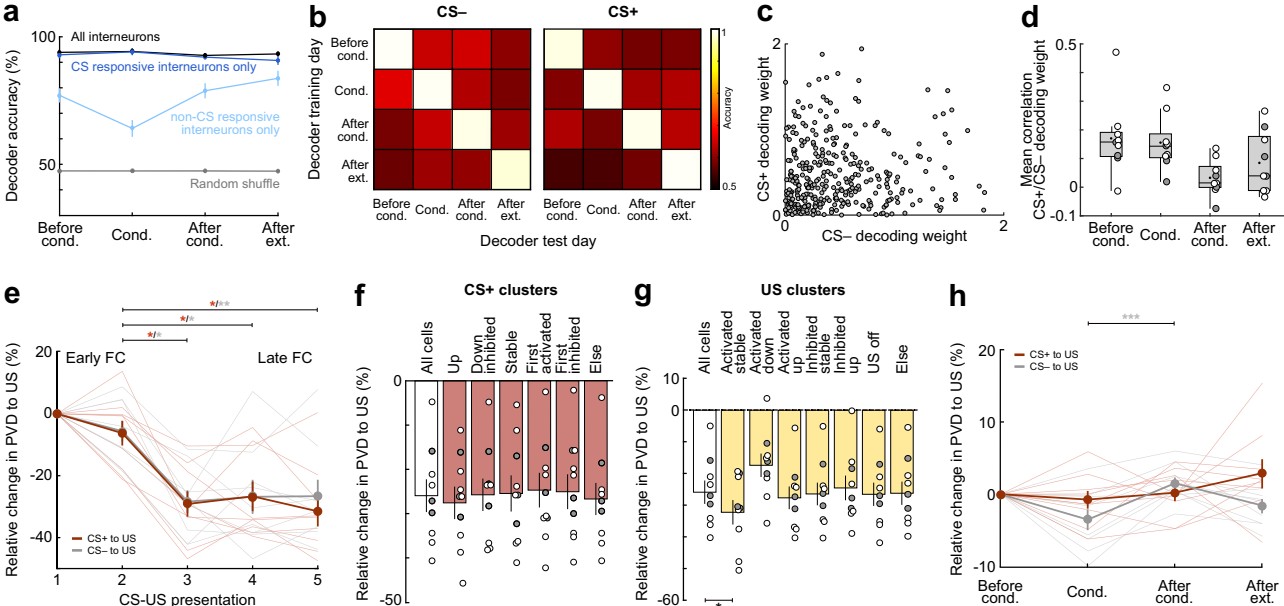

**Fig. 7 | Interneuron population activity dynamically changes day to day without losing overall stimulus representation. a** Mean intra-day decoder accuracy classifying CS+, CS− and baseline ($N = 9$ mice, $n = 100$ iterations), including accuracy using only CS-responsive cells, non-CS-responsive cells and shuffled labels. **b** Accuracy of inter-day two-way decoders classifying CS−/baseline (left) and CS+/baseline (right) ($N = 9$). **c** Example scatterplot of absolute CS+ and CS − decoding weights of each neuron in one iteration ($N = 9$ mice, $n = 333$ cells). **d** Mean correlation between absolute CS+ and CS− decoding weights per session ($N = 9$). **e** PVD change between CS+/US or CS−/US during conditioning ($N = 9$). CS +/US distance Friedman test ($\chi 2 = 13.9$), $p = 0.0030$, followed by Dunn's multiple comparisons (pairing 2 vs. 3, $p = 0.0115$; pairing 2 vs. 4, $p = 0.0209$; pairing 2 vs. 5, $p = 0.0115$). CS−/US distance Friedman test ($\chi 2 = 14.2$), $p = 0.0026$, followed by Dunn's multiple comparisons (pairing 2 vs. 3, $p = 0.023$; pairing 2 vs. 4, $p = 0.047$; pairing 2 vs. 5, $p = 0.023$). **f** Mean PVD change between early and late conditioning

($N = 9$), contribution of each CS+ activity pattern cluster was calculated by removing its cells (see Fig. 4). Friedman test ($\chi 2 = 13.9$), $p = 0.0304$, followed by Dunn's multiple comparisons (non-significant). **g** Mean PVD change between early and late conditioning for US clusters (see Fig. 3; $N = 9$). Friedman test ($\chi 2 = 36.0$), $p < 0.0001$, followed by Dunn's multiple comparisons ('All cells' vs. 'Activated stable', $p = 0.0272$). **h** Same as e but calculated across days ($N = 9$). CS−/US distance Friedman test ($\chi 2 = 14.2$), $p < 0.0001$, followed by Dunn's multiple comparisons ('Conditioning' vs. 'After conditioning', $p = 0.0005$). Data are presented as mean with s.e.m.; except (**d**) showing Tukey box-and-whisker plots with median, 25th and 75th percentiles, min to max whiskers with exception of outliers, dots indicate mean. Circles (**d**, **f**, **g**) represent individual animals (open circles, imaging sites in basal amygdala, $N = 6$; filled circles, border of lateral/basal amygdala, $N = 3$). Semi-transparent lines (**e**, **h**) represent individual animals. *$p < 0.05$, **$p < 0.01$, ***$p < 0.001$. See Supplementary Table 1 for statistics details.

in CS− cluster distribution could be observed between the two interneuron types (Fig. 8j). Overall, this data demonstrates that interneuron subpopulations are highly diverse even within a molecular subpopulation, with heterogeneous plasticity patterns visible for both activated and inhibited neurons. However, our analysis suggests that certain activity patterns are enriched in interneuron subpopulations. For example, given the high fraction of stable CS+ activated cells in the VIP interneurons, the noticeable CS+ responses we observed in the general BLA interneuron population during fear learning (Fig. 4) could be strongly driven by these cells.

Finally, we compared SST and VIP activity patterns across fear learning and extinction (SST, $n = 114$ cells from $N = 4$ mice; VIP, $n = 152$, $N = 6$). Like the conditioning day, differential responses between the subpopulations were mainly visible for the conditioned CS+ tone, and less for the CS− control tone. On the population level, SST interneurons were initially predominantly inhibited during the habituation session, however, upon extinction showed increased CS+ responses (Fig. 9a–c). In contrast, VIP interneurons were activated during habituation and after conditioning but showed suppressed activity after extinction. In comparison, this led to a stronger signalling of VIP interneurons upon neutral−but novel−tone presentations in habituation, while SST interneurons dominated after extinction (Fig. 9a–c). Analysis of the fractions of significantly modulated cells demonstrated that these differences are driven by suppression of SST interneuron activity before and after conditioning, since these cells showed significantly higher fractions of inhibited neurons (Fig. 9d). In contrast, after extinction, SST interneurons displayed more excitatory responses and a reduced fraction of unresponsive cells. This effect was not

detectable for the CS− control tone. Only after conditioning, SST interneurons showed more inhibition and VIP cells more activation to the CS− (Fig. S12a–d). No differences in the number of cells significantly modulated by the CS− were detected between the two analysed interneuron subpopulations over the course of learning and extinction (Fig. S12d). For both the CS+ and the CS−, clustering of CS responses revealed a higher proportion of extinction activated neurons in SST interneurons (SST 13%, VIP 7%) but a higher proportion of VIP cells stably activated during the entire paradigm (SST 12%, VIP 27%; Fig. 9e and Fig. S12e–i). While none of these disparities between the two subpopulations were found to be statistically different for the CS+, the proportion of VIP neurons stably encoding the CS− across days was significantly higher. Similar to the general interneuron population, CS+ cluster responses developed gradually across the two extinction sessions in SST and VIP interneurons (Fig. S13a, b). Clustering of CS+ responses for the entire extinction protocol (jointly for extinction 1 and 2) revealed a significantly higher proportion of VIP neurons in the 'Extinction 1 activated' cluster (Fig. S13c–e), consistent with the finding that this subtype shows a stronger activation after conditioning.

Lastly, we analysed the dynamics of SST and VIP interneurons during spontaneous immobility bouts outside of any auditory cues, in order to determine whether distinct subtypes of interneurons track internal fear states. SST interneurons were robustly inhibited during immobility epochs and reactivated at the onset of movement (Fig. 9f, h), which is consistent with an involvement in state transitions and fear suppression. In contrast, VIP interneurons showed overall no modulation during spontaneous freezing onset or offset (Fig. 9g, h). Taken

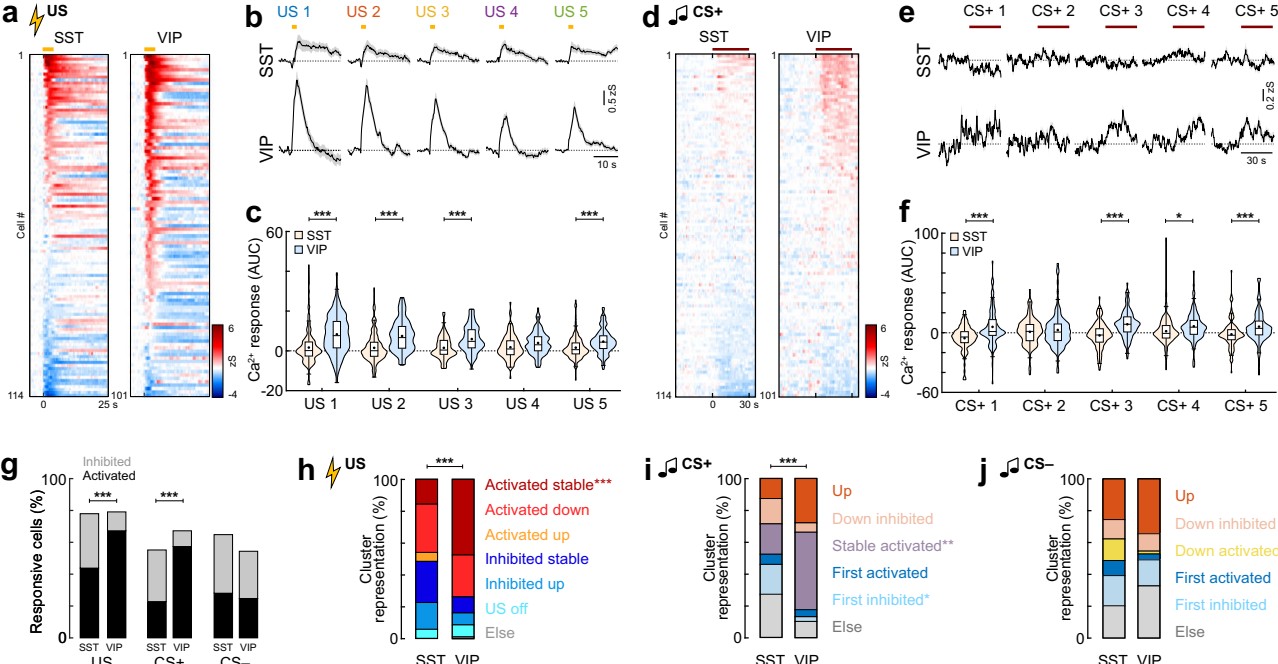

**Fig. 8 | Differential activity during fear learning in molecular interneuron subpopulations. a** Heatmap of US responses, sorted by amplitude (SST, $n = 114$ cells from $N = 4$ mice; VIP, $n = 101$, $N = 4$). Yellow line indicates US. **b** Average US responses (SST, $n = 114$; VIP, $n = 101$). **c** Area under the curve (AUC) during the US. Mann–Whitney test with Bonferroni correction, SST vs. VIP; US 1, $p < 0.0001$; US 2, $p < 0.0001$; US 3, $p < 0.0001$; US 5, $p = 0.0005$. **d** Heatmap of CS+ responses, sorted by individual amplitude (SST, $n = 114$ cells; VIP, $n = 101$). Line indicates CS + . **e** Average CS+ responses (SST, $n = 114$; VIP, $n = 101$). **f** AUC during CS+ (SST, $n = 114$; VIP, $n = 101$). Mann–Whitney test with Bonferroni correction; SST vs. VIP; CS+ 1, $p < 0.0001$; CS+ 3, $p < 0.0001$; CS+ 4, $p = 0.0287$; CS+ 5, $p = 0.0003$. **g** Proportions of responsive neurons during conditioning (SST, $n = 114$; VIP, $n = 101$). CS+, Chi-Square test ($\chi2(2) = 30.885$), $p < 0.0001$; SST vs. VIP, post hoc Chi-Square test with Bonferroni correction, 'Activated', $p < 0.0001$; 'Inhibited', $p = 0.0004$; US, Chi-

Square test ($\chi2(2) = 16.663$), $p < 0.0001$; SST vs. VIP, post hoc Chi-Square test with Bonferroni correction, 'Activated', $p = 0.0028$; 'Inhibited', $p = 0.0007$. **h** Proportions of cells in US clusters (SST, $n = 89$; VIP, $n = 80$). Chi-Square test ($\chi2(6) = 28.635$), $p < 0.0001$; SST vs. VIP, post hoc Chi-Square test with Bonferroni correction, 'Activated stable', $p = 0.0001$. **i** Proportion of cells in CS+ clusters (SST, $n = 63$; VIP, $n = 68$). Chi-Square test SST vs. VIP ($\chi2(5) = 28.155$), $p < 0.0001$; post hoc Chi-Square test with Bonferroni correction; 'Stable activated', $p = 0.0046$; 'First inhibited', $p = 0.0418$. **j** Proportion of cells in CS− clusters (SST, $n = 74$; VIP, $n = 55$). Average traces (**b**, **e**) are mean with s.e.m.; violin plots (**c**, **f**) show distribution of all data points; Tukey box-and-whisker plots median, 25th and 75th percentiles, min to max whiskers with the exception of outliers, dots indicate mean. $*p < 0.05$, $**p < 0.01$, $***p < 0.001$. See Supplementary Table 1 for statistics details.

together, our data demonstrate a stronger activation of VIP interneurons to external novel stimuli and in conditioned high fear states (CS before and after conditioning), but no modulation by internal fear states, suggesting that these interneurons are mediating cue-driven disinhibition to salient stimuli—likely via the VIP → SST → PN circuit motif. In contrast, BLA SST interneuron activity contributes to extinction-related inhibition and additionally encodes internal behavioural states, indicating that these cells could preferentially signal safety conditions. However, despite average preferences, a detailed analysis of response patterns showed that all CS response types can be detected in both molecular subpopulations. This was also reflected in a bimodal distribution of CS response magnitudes in individual SST interneurons after fear extinction (Fig. 9c), indicating the presence of further subtypes and potentially functionally distinct, state-dependent canonical microcircuits within these classical interneuron subpopulations.

## Discussion

Here, we used deep-brain imaging to follow large populations of amygdala interneurons at single cell resolution across days and provide a classification of their response types during fear learning, memory expression and extinction. We report that similar to neighbouring PNs[26,33,40], BLA interneurons develop complex activity patterns with plastic changes across associative fear learning and extinction. This plasticity was seen both at the level of individual cells and neuronal population coding, and differed for distinct molecular interneuron subpopulations.

## Encoding of high and low fear states in amygdala interneurons

At the single-cell level, BLA interneurons most prominently responded to the instructive US foot shock, but also to the predictive CS+ and control CS− tones during conditioning (Fig. 2). Analysis of the population average of BLA interneurons suggested a strong activation that declined over the course of repeated US presentations and thus predictive learning. At the same time, CS+ but not CS− responses on average increased in BLA interneurons. However, clustering of individual neuronal responses revealed highly diverse cellular responses beyond uniform activation. BLA interneurons showed plastic responses to both the predictive CS+ and the control CS−. Yet, CS+ and CS− functional clusters differed within the general inhibitory population—cells stably activated across all five presentations were selectively detected during the CS+, while a cluster of activated neurons that decreased their responses over the trials was only present during the CS−. This suggests that the overall increased CS+ response in the interneuron population was not simply caused by an upregulation of individual activity across trials, but was the result of stable activation of a subset of interneurons throughout the session, while CS− clusters displayed balanced up- and downregulation.

Across fear learning and extinction, amygdala interneurons showed similar CS activity patterns as previously described for PNs[26,33,40], such as selectively increased activity before conditioning, after fear learning and after extinction (Fig. 5). For each of these categories, we also found interneuron clusters that were significantly inhibited by the CS. Since amygdala interneurons are tonically active in vivo, leading to persistent inhibition of downstream PNs or

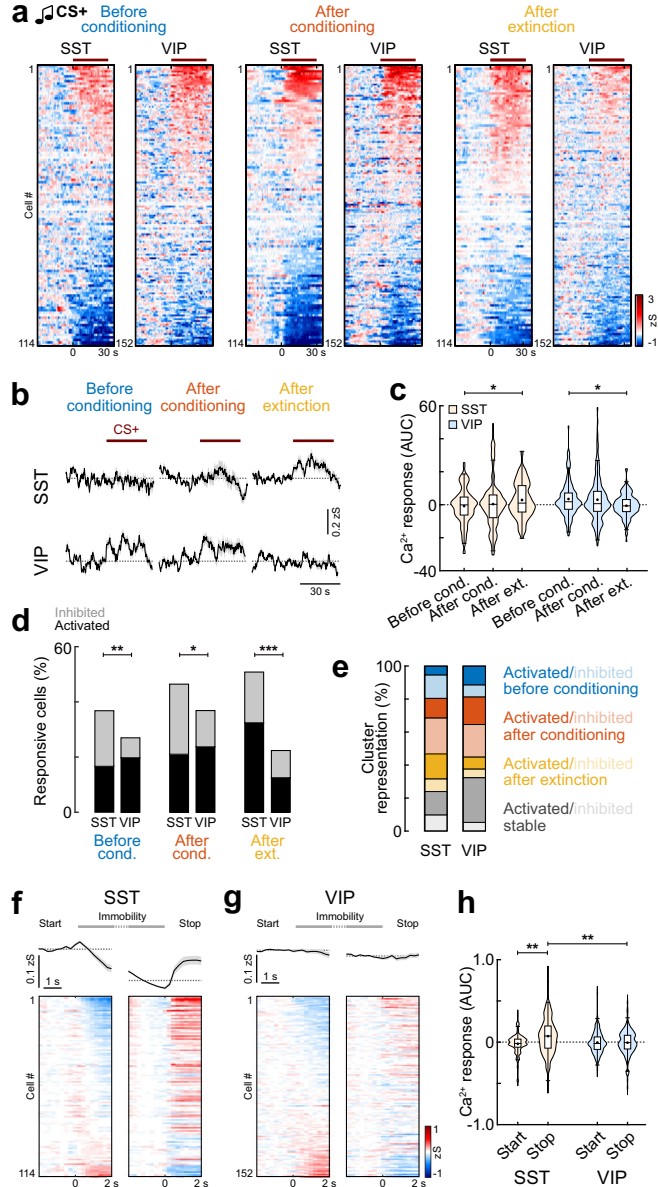

**Fig. 9 | Interneuron subpopulations contribute differently to the encoding of fear states. a** Heatmap of SST and VIP BLA interneuron responses to the CS+ before conditioning, after conditioning and after extinction, sorted by individual response amplitude (SST, $n = 114$ cells from $N = 4$ mice; VIP, $n = 152$, $N = 6$). Line indicates CS duration. **b** Corresponding average CS+ responses (SST, $n = 114$; VIP, $n = 152$). **c** Area under the curve (AUC) during the CS+ (SST, $n = 114$; VIP, $n = 152$). Paired Wilcoxon test with Bonferroni correction; SST 'Before conditioning' vs. 'After extinction', $p = 0.036$, VIP 'Before conditioning' vs. 'After extinction', $p = 0.012$. **d** Proportions of responsive neurons (SST, $n = 114$; VIP, $n = 152$). 'Before conditioning', Chi-Square test ($\chi2(2) = 9.7873$), $p = 0.0075$, SST vs. VIP, post hoc Chi-Square test with Bonferroni correction, 'Inhibited', $p = 0.0098$; 'After conditioning', Chi-Square test ($\chi2(2) = 6.5609$), $p = 0.0376$, SST vs. VIP, post hoc Chi-Square test with Bonferroni correction, 'Inhibited', $p = 0.0496$; 'After extinction', Chi-Square test ($\chi2(2) = 23.938$), $p < 0.0001$, SST vs. VIP, post hoc Chi-Square test with Bonferroni correction, 'Activated', $p = 0.0005$, 'No response', $p < 0.0001$. **e** Proportion of cells in CS+ clusters (SST, $n = 92$; VIP, $n = 96$). **f** Average traces and heatmap of SST interneuron activity during immobility bouts (Test/Extinction 1 and Extinction 2), aligned to immobility start and movement onset, respectively, sorted by amplitude upon start ($n = 114$). **g** Same for VIP interneurons ($n = 152$). **h** Corresponding AUC for immobility start and stop (2 s duration). Kruskal Wallis test ($H = 15.33$), $p = 0.0016$ with post hoc comparison, SST start vs. SST stop, $p = 0.0025$; SST stop vs. VIP stop, $p = 0.0177$. Average traces (**b, f, g**) are mean with s.e.m.; violin plots (**c, h**) show distribution of all data points; Tukey box-and-whisker plots median, 25th and 75th percentiles, min to max whiskers with exception of outliers, dots indicate mean. *$p < 0.05$, **$p < 0.01$, ***$p < 0.001$. See Supplementary Table 1 for statistics details.

interneurons[15,19,25], suppression of their activity after learning can induce selective disinhibition, allowing for circuit computations necessary for further learning and memory expression. This inhibitory plasticity can, for example, stabilise memory traces or increase the selectivity of engrams[41], or enhance the contrast between distinct long-range PN circuits associated with fear or extinction states[23,32]. Unlike BLA PNs[26,40], we could not detect an extinction-resistant activity pattern in interneurons that reflects a persistent encoding of the original fear memory, even after extinction[42]. By contrast, interneurons exhibit more dynamic, state-dependent activity patterns, suggesting that they regulate the moment-to-moment expression of fear or safety rather than storing long-term associations. This functional distinction may point to a fundamental organisational principle of fear circuits: PNs act as stable encoders of learned associations, while interneurons provide flexible inhibitory gating of PN output and plasticity, while probably keeping selectivity as to which PN subpopulations (e.g., fear or extinction neurons) they are connected to. This enables adaptive switching between fear and safety states, and updating of learned information or higher order learning, likely through the integration of contextual and top-down inputs. This division supports the notion that fear memories are not erased during extinction, but rather suppressed through inhibitory circuit mechanisms, and underscores the pivotal role of local interneurons in facilitating behavioural flexibility.

Our findings demonstrate that subsets of BLA interneurons activated by conditioned auditory cues are also modulated during internally generated fear states, in the absence of external sensory input. Specifically, we observed that 'Fear inhibited' and, to a lesser extent, 'Extinction activated' interneurons were consistently suppressed during spontaneous freezing and reactivated upon movement onset, exhibiting stable activity patterns across days. These dynamics reflect their responses to cues, i.e. suppression during the presentation of the conditioned stimulus in the context of fear, and activation during extinction. This suggests that these cell types encode both stimulus-bound associations and more generalised internal fear or safety states. Our data support the idea that the BLA contains inhibitory ensembles that bridge the sensory-driven and state-dependent aspects of fear regulation. 'Fear inhibited' interneurons typically restrain PN activity, but are temporarily silenced to allow fear expression by disinhibiting PN ensembles that encode the CS+ and/or mediate freezing behaviour, thereby enabling a robust output from the BLA to downstream targets. The reactivation of these interneurons at the offset of freezing further implies their involvement in terminating fear responses and restoring inhibitory control. Together, these dynamics suggest that this interneuron population gates fear expression by modulating PN excitability and state-dependent plasticity rather than encoding associative information per se.

When examining the relationship between interneuron recruitment and behavioural performance, we found that mice with a higher fraction of CS+ 'Fear inhibited' neurons exhibited significantly better discrimination between CS+ and CS−, which is consistent with the idea that transient disinhibition during the expression of fear supports the accurate encoding of threats. Conversely, mice with more 'Extinction activated' neurons tended to discriminate more poorly. This may reflect a pre-existing inhibitory bias that limits fear expression in animals with less precise associative memory. These findings highlight a potential inhibitory circuit-level substrate for variability in fear discrimination and generalisation between individuals. In contrast, 'Fear activated' and 'Extinction inhibited' interneurons exhibited minimal modulation during spontaneous freezing and were not correlated with behavioural performance, despite robust cue-related responses. This suggests that these cells primarily encode stimulus-specific associations without directly influencing the balance between generalisation and discrimination. Overall, our results indicate a functional segregation within BLA interneuron populations: some encode sensory-specific associative information, while others integrate sensory

inputs with dynamic internal states to influence fear expression and extinction.

## Functional divergence of amygdala interneuron subtypes in stimulus and state encoding

Some of the activity patterns we observed during distinct external cues or in discrete behavioural states in the general inhibitory BLA population were enriched in SST and VIP interneurons. VIP interneurons were overall more excited by novel auditory cues during habituation, and displayed overall stronger CS+ responses compared to SST interneurons during the progression of fear learning. VIP interneurons showed a significantly larger fraction of stably activated neurons and more upregulation of CS+ responses during conditioning, while SST interneurons had a comparably higher fraction of CS+ inhibited cells. This is consistent with the notion that PN dendritic disinhibition via VIP-mediated inhibition of SST cells supports sensory processing during associative fear learning[15,16,19]. Increased activity in VIP interneurons, which enables such disinhibition, might facilitate the processing of novel and salient information in microcircuits, as recently also shown in other brain areas[43–45]. In the amygdala, elevated cue-related VIP activity in high fear states could similarly enable the integration of context-dependent information or promote higher-order conditioning. In contrast, SST interneurons were predominantly activated in low fear states after extinction, which was recently also reported for hippocampal SST cells[46]. Our findings that SST interneurons preferentially signal safety in fear learning are consistent with previous studies, showing increased BLA SST interneuron activity to learned non-threatening cues[18]. In associative fear conditioning, increased activity of dendrite-targeting SST interneurons would perturb processing of inputs at PN dendrites and thus suppress PN responsiveness to auditory cues, which would represent a mechanism to suppress fear neuron activity[40]. To test whether these subtype-specific responses extend beyond stimulus-bound activation to internal fear states, we examined SST and VIP interneuron activity during spontaneous immobility bouts in the absence of auditory cues (Fig. 9). SST interneurons were largely inhibited at freezing onset and reactivated upon movement resumption, matching previous reports in hippocampal circuits[46], and suggesting an additional role in tracking behavioural state transitions and facilitating safety-related processing. In contrast, VIP interneurons showed no clear modulation during spontaneous freezing, which is consistent with the idea that they are specialised for external, stimulus-driven disinhibition rather than internal state representation.

These findings emphasise functional differences between BLA interneuron subtypes. VIP interneurons appear to be tuned to salient external sensory cues, particularly during the acquisition and expression of aversive associations, while SST interneurons—in addition to mediating cue-related dendritic disinhibition—track internal behavioural states and are also suppressed in high fear during immobility, which might indicate a multiplexed role for sensory- and state-dependent gating of PN plasticity. These results suggest that the canonical VIP → SST → PN disinhibitory motif functions in a dynamic, state-dependent manner in the BLA, presumably shaped by distinct long-range inputs. Activation of VIP interneurons to external aversive and auditory stimuli is likely driven by excitatory inputs from the sensory cortex and thalamus[19,47]. This facilitates fear learning and expression by disinhibiting PN dendrites via the suppression of SST interneurons. Conversely, SST interneuron activation in low fear states in extinction may be triggered by inputs from the hippocampus and prefrontal cortex[18,19,48], thereby providing feedforward inhibition to PN dendrites and suppressing fear expression. Although their differential recruitment by external inputs is likely a key driver of experience- and state-dependent activity, the reciprocal inhibitory connectivity between BLA SST and VIP interneurons[19] might reinforce and shape their inverse population dynamics during fear expression and

extinction. In parallel, competition between functionally distinct ensembles of excitatory neurons supported by local inhibitory elements—such as PNs encoding fear and extinction[23,32,40]—may further bias the network's output towards either expressing or suppressing fear, thereby reinforcing state-dependent microcircuit adaptations. Notably, although all response types were generally represented within both molecular subtypes, SST and VIP interneurons showed distinct biases in their activity profiles, suggesting that molecular identity does not strictly define function, but rather shapes the relative contribution of each subtype to provide a weighted output signal for circuit-level computations in fear and extinction. Moreover, this diversity might reflect the existence of separate high and low fear state canonical BLA microcircuits. Overall, our findings suggest that functional biases of interneuron subtypes enable flexible modulation of BLA output, depending on the behavioural context and learning stage.

## Mechanisms of interneuron plasticity

The question remains as to where this plasticity is located—are interneurons and their synapses indeed undergoing plastic changes themselves, or are changes in activity patterns across learning and extinction simply imposed by synaptic inputs? Both local PNs[16,26,33] and long-range excitatory afferents e.g. from auditory thalamus[35,47] display similar activity patterns in associative learning as observed here for amygdala interneurons. Since they have been shown to impinge on several BLA interneuron subpopulations[19,24,49], auditory thalamus or local PN inputs could relay plastic activity patterns, which would additionally lead to feed-forward inhibition of other interneurons within the interconnected BLA microcircuits.

In addition, there is ample evidence from biochemical, electrophysiological and anatomical ex vivo studies to support the idea of local plasticity within BLA interneurons. For example, aversive learning alters the expression of enzymes for GABA synthesis, GABA receptors and their scaffolding protein gephyrin in the BLA[3–5]. Glutamatergic inputs to amygdala interneurons can be potentiated by tetanic stimulation, which leads to an increase of inhibitory synaptic drive onto PNs[6–8]. Conversely, fear learning can modulate excitatory inputs onto distinct subpopulations of amygdala interneurons, such as PV and cholecystokinin (CCK) cells[21,50]. Together, these in vitro data point to a bidirectional regulation of interneuron plasticity upon associative learning, which is in line with our observations of cells that are selectively activated or inhibited in fear expression. Furthermore, fear conditioning has been shown to induce structural remodelling of GABAergic synapses that can be reversed by extinction[51]. However, extinction does not simply reverse conditioning-induced potentiation[51,52], which would resemble a process of forgetting. Considered to be a new form of context-dependent safety learning that suppresses the original fear memory[53], extinction has additionally been associated with potentiation of excitatory synapses on inhibitory interneurons, but also GABAergic synapses on PNs[10,12]. This is reflected in extinction-mediated up- and downregulation of CS responses in distinct subsets of interneurons in the present study. Altogether, our in vivo results are therefore consistent with previous ex vivo work on cellular mechanisms of associative fear learning and extinction.

Our data offer additional insights into behaviourally-mediated plasticity of individual inhibitory BLA neurons. Interneurons displayed overlap of sensory representations during learning, with cells encoding the CS+, CS− and US being spatially intermingled in the BLA (Figs. 2 and 7c, d). As this overlap was not found to be greater than would be expected by chance, it appears that auditory and aversive information remain largely segregated during fear conditioning, with no immediate non-linear integration at the level of inhibitory interneurons. This suggests that their integration may require additional network effects during consolidation, including local PNs or upstream brain regions. Further, a large proportion of interneurons was activated by both the predictive CS+ and the instructive US, making them

ideal candidates for cellular plasticity during Hebbian learning. However, upregulation of CS+ responses in individual cells during fear learning (Fig. 4) or expression (Fig. 5) was independent of their US activation during conditioning, suggesting that somatic CS–US coincidence is not necessary for learning-associated adaptation of interneuron activity. Yet, our somatic recordings cannot capture converging CS and US inputs in dendrites, which could induce dendritic plasticity in interneurons[54–56] and thus represent a potential source of CS response enhancement. Moreover, similar proportions of interneurons down-regulated their CS response upon fear learning, including cells that were significantly activated by the US. This supports the idea that memory processes involve diverse forms of plasticity beyond classical Hebbian learning in interneurons, as previously proposed for amygdala PNs[16,33].

### Ensemble coding in amygdala interneurons

At the population level, we found that CS+, CS− and baseline could be reliably decoded from interneuron activity within each experimental day, but not across behavioural states (Fig. 7). This suggests that CS information is stably encoded within the ensemble, although the activity of individual interneurons changes dynamically from day to day during learning and extinction. Decoding accuracy was highest when all interneurons or only 'CS responsive' cells were included in the sample, while decoders trained on 'non-CS responsive' interneurons exhibited reduced accuracy (although, still higher than randomly shuffled data). This might be due our strict definition of a responsive neuron, which is based on at least three significant responses during four CS+ or CS− presentations, respectively. The higher-than-random accuracy for 'non-CS responsive' interneurons suggests that these cells indeed carry valuable information about the auditory cues that elude our strict definition of 'responsive'.

Using population vector distance analysis, we could further observe that the representation of both the CS+ and the CS− was getting more similar to the aversive US during learning. In contrast to BLA PNs, where this effect is selective for CS+ encoding and mainly mediated by up- and downregulation of CS responses[33], we found that defined individual CS+ clusters of interneurons contributed little to this effect. Instead, the reduction in the population vector distance between CS+ and the aversive US signal seems to be driven by interneurons across all clusters, yet can be significantly affected by the 'Activated stable' US cluster. This effect might be mediated by VIP interneurons, since this cluster is overrepresented in this interneuron subpopulation (Fig. 8). Moreover, unlike the plasticity at the single neuron level, learning-induced changes at the population level during conditioning were transient and did not persist over days. This suggests that although representations of sensory stimuli change during fear learning, unlike PNs[33], this transient shift does not consolidate overnight, which might ensure valence-free representations of environmental cues within the BLA interneuron population. Additional studies will be needed to address how individual interneuron subpopulations contribute to these high-dimensional representations.

### Implications of functional interneuron diversity

Remarkably, across days and stimuli, amygdala interneurons displayed high diversity in their response patterns, even within a molecular subpopulation. Population averages were dominated by activated neurons even when similar proportions of cells were classified as activated or inhibited (see e.g. Fig. 3). This is likely due to asymmetric effects of suppression or increase of neuronal spiking, which will strongly depend on the baseline firing rate of any given neuron, but also individual differences in calcium buffering as well as non-linearity of GCaMP indicators[57]. Together, these results highlight that any interpretation that a molecularly-defined interneuron group is homogeneously activated based on population averages (e.g. fibre-photometry recordings) should be made with caution. Similarly, opto- or chemogenetic manipulations that uniformly drive the entire population of any given interneuron subpopulation will artificially overrule the physiologically diverse response patterns during behaviour[58].

But how can one make sense of this functional diversity of interneurons? Non-uniform responses across a neuronal population of amygdala neurons have behavioural advantages, as they enable for example avoidance of generalisation by selective and precise computations to specific environmental cues, and allows for circuit adaptations in dependence of varying internal states[26,59,60]. However, the broad molecular subpopulations currently used to classify interneurons obscure the diverse molecular, morphological, physiological and functional characteristics of individual cells. A true definition of an interneuron subtype must be based on its functional properties and connectivity, i.e. its role in a neural circuit. Molecular classification provides genetic entry points to interneuron targeting and might correlate with other cellular properties, but the currently employed genetic mouse lines, such as SST-Cre or VIP-Cre, might not be sufficient to account for the reported diversity in morphology, connectivity and function at the single-cell level[15,19,24]. Indeed, a recent study on cortical SST interneurons has shown selective morphology and connectivity of SST molecular subclasses[61], and similar observations have been made for cortical and hippocampal VIP interneurons[62,63]. Similarly, in the BLA, VIP interneurons targeting other interneurons vs. those connecting to PNs can be identified by co-expression patterns with cholecystokinin[20]. Recent advances in single-cell RNA sequencing now enable an even finer grained interneuron classification in the BLA[64–66], allowing to target further subclasses with intersectional genetic approaches. Yet, these are limited to two or three genes[67], and will greatly reduce cell numbers that can be recorded simultaneously and thus affect the interpretability of results. In the future, novel high-plex spatial transcriptomics after unbiased in vivo functional recordings[68–70] can help to determine whether discrete molecular and anatomical characteristics of interneurons also correlate with distinct functional properties.

## Methods

### Animals

All animal procedures were performed in accordance with institutional guidelines at the Friedrich Miescher Institute for Biomedical Research and were approved by the Veterinary Department of the Canton of Basel-Stadt. Heterozygous (cre/wt) GAD2-Ires-Cre, VIP-Ires-Cre and SST-Ires-Cre mice[28] fully backcrossed to a C57BL/6J background were used for virally mediated, Cre-dependent expression of a calcium indicator. Experiments were performed with male (GAD2-Cre, VIP-Cre, SST-Cre) and female (GAD2-Cre) mice aged 2–3 months at the time of injection. Animals were kept in a 12 h light/dark cycle (22–24 °C, 40–60% humidity) with access to food and water ad libitum and were individually housed after implant surgeries. All experiments were conducted during the light cycle. Of note, a different analysis on fear conditioning day data from VIP-Cre and SST-Cre mice of this study was previously published[19].

### Surgical procedures

Surgical procedures were performed as previously described[19,26]. In brief, mice were anaesthetised using isoflurane (3–5% for induction, 1–2% for maintenance; Attane, Provet) in oxygen-enriched air (Oxymat 3, Weinmann) and fixed on a stereotactic frame (Model 1900, Kopf Instruments). Injections of buprenorphine (Temgesic, Indivior UK Limited; 0.1 mg/kg body weight subcutaneously 30 min prior to anaesthesia) and ropivacaine (Naropin, AstraZeneca; 0.1 ml locally under the scalp prior to incision) were provided for analgesia. Postoperative pain medication included buprenorphine (0.1 mg/kg in the drinking water; overnight) and injections of meloxicam (Metacam, Boehringer Ingelheim; 1 mg/kg subcutaneously) for up to three days if necessary. Ophthalmic ointment (Viscotears, Bausch and Lomb) was

applied to avoid eye drying. Body temperature of the experimental animal was maintained at 36 °C using a feedback-controlled heating pad. AAV2/9.CAG.flex.GCaMP6f or AAV2/9.CAG.flex.GCaMP6s[27] (400 nl, University of Pennsylvania Vector Core, UPenn) was unilaterally injected into the BLA using a precision micropositioner (Model 2650, Kopf Instruments) and pulled glass pipettes (tip diameter about 20 μm) connected to a Picospritzer III microinjection system (Parker Hannifin Corporation) at the following coordinates from bregma: AP −1.5 mm, ML −3.3 mm, DV 4.1–4.5 mm below the cortical surface. The skin incision was closed with polypropylene suture (Prolene 6-0, Ethicon) and the animal placed into a recovery cage on a heating pad until fully mobile. Two weeks after virus injection, a gradient-index microendoscope (GRIN lens, 0.6 × 7.3 mm, GLP-0673, Inscopix) was implanted into the BLA using the same surgical approach. A sterile needle (0.7 mm diameter) was used to make an incision above the implant site. The GRIN lens was subsequently lowered into the brain with a micropositioner (coordinates from bregma: AP −1.6 mm, ML −3.2 mm, DV 4.5 mm below the cortical surface) using a custom-build lens holder and fixed to the skull using UV light-curable glue (Loctite 4305, Henkel). The skull was sealed with Scotchbond (3 M), Vetbond (3 M) and finally dental acrylic (Paladur, Heraeus). A custom-made head bar for animal fixation during the miniature microscope mounting procedure was embedded into the dental cement. Mice were allowed to recover for at least one week after GRIN lens implantation before starting to check for GCaMP expression.

### Deep-brain calcium imaging
Starting one week after GRIN lens implantation, mice were head-fixed to check for sufficient expression of GCaMP using a miniature microscope (nVista HD, Inscopix). Two to four weeks after the implant surgery, mice were briefly anaesthetised with isoflurane to fix the microscope baseplate (BLP-2, Inscopix) to the skull using light-curable composite (Vertise Flow, Kerr). The microscope was removed, and the baseplate capped with a baseplate cover (Inscopix) whenever the animal was returned to its home cage. The microscope was mounted daily immediately before starting the behavioural session. Mice were habituated to the brief head-fixation on a running wheel for miniature microscope mounting for at least three days before the behavioural paradigm. Imaging data was acquired using nVista HD 2.1 software (Inscopix) at a frame rate of 20 Hz with an LED power of 40-80% (0.9−1.7 mW at the objective, 475 nm), analogue gain of 1–2 and a field of view of 650 × 650 μm. For individual mice, the same imaging parameters were kept across repeated behavioural sessions.

### Behaviour
Two different contexts were used for the associative fear learning paradigm. Context A (retrieval context) consisted of a clear cylindrical chamber (diameter: 23 cm) with a smooth floor, placed into a dark-walled sound attenuating chamber under dim light conditions (approximately 25 lux). The chamber was cleaned with 1% acetic acid. Context B (fear conditioning context) contained a clear square chamber (26 × 26 cm) with an electrical grid floor (Coulbourn Instruments) for foot shock delivery, placed into a light-coloured sound attenuating chamber with bright light conditions (approximately 180 lux), and was cleaned with 70% ethanol. Both chambers contained overhead speakers for delivery of auditory stimuli, which were generated using a System 3 RP2.1 real-time processor and SA1 stereo amplifier with RPvdsEx 78 software (all Tucker-Davis Technologies). A precision animal shocker (H13-15, Coulbourn Instruments) was used for the delivery of alternating current (AC) foot shocks through the grid floor. Behavioural protocols for stimulus control were generated with Radiant 2.0 Software (Plexon) via TTL pulses. On day 1, mice were habituated in context A. Two different pure tones (conditioned stimulus, CS; 6 kHz and 12 kHz, total duration of 30 s, consisting of 200 ms pips repeated at 0.9 Hz; 75 dB sound pressure level) were presented five times each in an alternated fashion with a pseudorandom ITI (range 60−90 s, 2 min baseline before first CS). On day 2, mice were conditioned in context B to one of the pure tones (CS+) by pairing it with an unconditioned stimulus (US; 2 s foot shock, 0.65 mA AC; applied after the CS at the time of the next expected pip occurrence). The other pure tone was used as a CS− and not paired with a US. CS+ with US and CS− were presented alternating five times each in a pseudorandom fashion (ITI 60−90 s), starting with the CS+ after a 2 min baseline period. Animals remained in the context for 1 min after the last CS− presentation and were then returned to their home cage. On day 3 and 4, fear memory was tested, and extinction induced in context A. After a 2 min baseline period, the CS− was presented four times, followed by 12 CS+ presentations (ITI 60-90 s). The use of 6 kHz and 12 kHz as CS+ was counterbalanced across animals. Behavioural videos were acquired with an overhead camera and the Cineplex Studio 3.4.1 software (Plexon). Timestamps of calcium imaging frames, behavioural videos and external stimuli were collected for alignment on a master clock using the MAP 2.7 data acquisition system (Plexon).

### Histology
Mice were deeply anaesthetised with urethane (2 g/kg body weight; intraperitoneally) and transcardially perfused with 0.9% NaCl followed by 4% paraformaldehyde in PBS. The GRIN lens was removed and brains post-fixed in 4% paraformaldehyde for at least 2 h at 4 °C. Coronal sections (120 μm) containing the BLA were cut with a vibratome (VT1000S), immediately mounted on glass slides and cover-slipped using Vectashield (Vector Laboratories). To verify the GRIN lens position, sections were scanned with a laser scanning confocal microscope (LSM700, Carl Zeiss AG) equipped with a 10x air objective (Plan-Apochromat 10x/0.45) and matched against the Allen Mouse Brain Reference Atlas (https://mouse.brain-map.org).

A subset of animals was used for immunohistochemical analysis of interneuron marker gene expression in GCaMP6f+ neurons of *GAD2-Cre* mice. Here, brains were cut into 60 μm coronal slices with a vibratome. Sections were washed in PBS four times and blocked in 10% normal horse serum (NHS, Vector Laboratories) and 0.5% Triton X-100 (Sigma-Aldrich) in PBS for 2 h at room temperature. Slices were subsequently incubated in a combination of the following primary antibodies in carrier solution (1% NHS, 0.5% Triton X-100 in PBS) for 48 h at 4 °C: rabbit anti-VIP (1:1000, Immunostar, 20077, LOT# 1339001), guinea pig anti-VIP (1:500, Synaptic Systems, 443005, LOT# 3-11), rat anti-SST (1:500, Merck Millipore, MAB354, LOT# 232625, 3474070), guinea pig anti-PV (1:500, Synaptic Systems, 195004, LOT# 195004/10 and Synaptic Systems, 195308, LOT# 1-3, 1-9), rabbit anti-pro-CCK (1:500, Frontiers Institute, CCK-pro-Rb-Af350, LOT# 453), chicken anti GFP (1:1000, Thermo Fisher Scientific, A10262, LOT# 2738236). After washing three times with 0.1% Triton X-100 in PBS, sections were incubated for 12−24 h at 4 °C with a combination of the following secondary antibodies in carrier solution: goat anti-chicken Alexa Fluor 488 (1:750, Thermo Fisher Scientific, A11039, Lot# 2420700), goat anti-rabbit Alexa Fluor 568 (1:750, Thermo Fisher Scientific, A11011, Lot# 2782620), goat anti-rabbit Alexa Fluor 647 (1:750, Thermo Fisher Scientific, A21245, Lot# 1778005), goat anti-rat Alexa Fluor 568 (1:750, Thermo Fisher Scientific, A11077, Lot# 692966, 2217022), goat anti-guinea pig Alexa Fluor 647 (1:750, Thermo Fisher Scientific, A21450, Lot# 2231672), goat anti-guinea pig DyLight 405 (1:250, Jackson ImmunoResearch, 106-475-003, Lot# 126016). Some sections were incubated in Hoechst 33258 (10 μg/ml, Thermo Fisher Scientific, H3569) for 10 min after washing out secondary antibodies with PBS to stain cell nuclei. After washing four times in PBS, sections were mounted on glass slides and cover-slipped with Vectashield. Sections were scanned using a laser scanning confocal microscope (LSM700) equipped with a 20× air objective (Plan-Apochromat 20×/0.8), or a Visitron VisiScope Spinning Disk confocal microscope (Visitron Systems GmbH) equipped with a 20× air objective (Plan-Apochromat 20×/0.8)

using 405 nm, 488 nm, 561 nm and 640 nm laser lines. Tiled z-stacks (3 μm step size) of the BLA were acquired and stitched with the Zeiss software processing tool (ZEN 2.3, black edition, Carl Zeiss AG) or the VisiView Software (VisiView5.0, Visitron Systems GmbH). Images were imported into Imaris software (Imaris 9.9.1, Bitplane) to count GCaMP+ somata and cells with co-expression of the peptide/protein of interest. Sections were inspected through the entire z-stack, and somata within the BLA borders were marked using the Imaris spot function. PV and SST were analysed in the same sections, and CCK and VIP together in adjacent sections, using every third section across the entire rostro-caudal extend of the BLA (4–5 sections per antibody combination for each mouse, $N = 3$ mice, $n = 541 \pm 53$ GCaMP+ cells for PV/SST, $n = 624 \pm 97$ cells for VIP/CCK).

### Calcium imaging analysis

**Pre-processing.** Raw image data was analysed as previously described[19,26]. In brief, videos were spatially down sampled (4×), bandpass filtered (Fourier transform) and normalised by the filtered image in ImageJ (2.0.0-rc-49/1.51a, NIH). The movies from all the sessions of each animal were concatenated into a single file and motion-corrected with the non-rigid algorithm NormCorre using the CaImAn package (1.9.13)[71]. Cell detection was carried out using Principal Component Analysis (PCA) and Independent Component Analysis (ICA) with the CIAtah package (4.5.9)[72]. ROIs were initially oversampled and then manually inspected. ROIs not matching individual neurons in the motion-corrected video were removed. After confirming the individual cell ROIs, a cell map was generated by aggregating the spatial footprints of all identified cells within the field of view. The selected ROIs were then used to extract the raw fluorescence traces using CIAtah based on the 20 Hz video. Fluorescence traces were z-scored and binned in 250 ms for all further analyses.

**Data analysis.** To analyse responses around events (e.g. tone presentation), z-scored traces were baselined to the period before the event. For CS presentations, the baseline was set to 30 s and for US presentations to 5 s. Then, to detect statistically responsive interneurons, we compared the fluorescence trace during the baseline period to the trace evoked during the event using the non-parametric Wilcoxon rank-sum test. Only cells that displayed statistically significant responses ($p < 0.01$) to at least 3 CS or US presentations were classified as 'responsive'. For across-day comparisons, the first four CS presentations during habituation were compared to the four CS− presentations during day 3 (test/extinction 1) and 4 (extinction 2), and the first and last four CS+ presentations for test and extinction sessions, respectively. This was done to compensate for the uneven number of CS+ and CS− presentations across habituation and test/extinction days.

To identify subclasses of responses, PCAs with K-means clustering were performed on the baselined traces of significantly responsive interneurons (using an explained variance of at least 80%). To evaluate cluster segmentation, we calculated the Silhouette coefficient for each cell, a metric that considers both intra-cluster cohesion and inter-cluster separation. The number of clusters from 2 to 20 was evaluated using MATLAB built in function *evalclusters*. Two clusters were found as the best division based on the mean Silhouette coefficient. This cluster segmentation clearly divided the data into activated and inhibited responses. However, we noticed that our data exhibited more heterogeneity than this activated/inhibited classification. Therefore, we manually increased the cluster number to better represent the diversity of response patterns, while maintaining a mean Silhouette coefficient close to the one obtained with two clusters. The number of clusters generated by K-means were therefore initially set to 16 for US response patterns and 8 for CS+ and CS− response patterns during conditioning. For across days, they were set to 10 for both CS+ and CS− response patterns. All activity clusters were visually inspected,

classified, and merged if necessary. To examine CS+ response dynamics across extinction, neuronal responses to the first, middle, and last blocks of four CS+ trials were averaged in each extinction session. The averaged responses from all cells in both extinction sessions were concatenated for PCA and K-means clustering analysis, which was performed with 5 clusters. The area under the curve (AUC) for any CS response was calculated for 30 s following tone onset, for US presentations 5 s after foot shock onset.

To assess spatial differences among responsive cells, we calculated pairwise Euclidean distances within subgroups of cells responding to the same stimulus (e.g. the US). For each cell, the within-group distance was defined as the mean pixel distance to all other cells responding to the same stimulus. Cumulative distributions of the within-group distance for each stimulus (CS+, CS− and US) are shown together.

To determine neuronal activity during freezing, fluorescence traces were aligned to the onset or offset of immobility bouts and baseline-corrected to the 2 s preceding the event. To avoid confounding tone responses, only events occurring outside any CS+ and CS− presentations were included. Traces from day 3 (test/extinction 1) and 4 (extinction 2) session were pooled together and averaged to obtain a single immobility trace per cell. To analyse stability of neuronal responses across days, neuronal activity in sessions on day 1 (habituation), day 3 and 4 were analysed separately. The AUC was calculated for 2 s following immobility onset or offset (=movement onset). Spearman correlations between immobility onset and offset mean responses were calculated across sessions for each animal and statistically tested across animals using one-sample t-tests.

For *VIP-Cre* mice, $N = 2$ animals were excluded from the conditioning day analysis as one of the five US exposures could not be verified with the behavioural recordings. However, these mice were included for across day analysis, since they learned the association between CS+ and US with the remaining 4 pairings.

**Population analysis.** To measure the similarity between two sets of neuronal ensemble response patterns, we calculated the population vector distance (PVD) between activity vectors as described in Taylor et al.[35]. The distance between CS and US activity vectors was calculated using 30 s binned responses to CS presentations and the mean 5 s response to the US. Activity vectors were of length $n$ interneurons, for example, a population vector for the US presentation was created based on the mean response of each interneuron to the foot shocks (e.g. for an animal with 40 cells, the vector would have 40 mean fluorescence values). The Euclidean distance between each CS bin vector and the mean US response vector was then calculated and averaged for each CS presentation. During fear conditioning, the change in distance between CS and US vectors was normalised to the PVD of the first CS/US pairing in the session. Negative percentages indicate that CS and US activity patterns are becoming more similar, while positive percentages indicate they are diverging. For the fear conditioning session, we furthermore calculated the difference between the mean change in PVD in the early stage of conditioning (pairings 1–2) and the late stage of conditioning (pairings 4–5). For across-days PVD analysis, the PVD in each day was the result of averaging the PVD from the first 4 CS presentations to the mean US vector from conditioning, and the change in PVD was normalised to the distance between CS and US during habituation day.

**Decoders.** The calcium traces to the CS presentations and baselines were used to train binary linear support vector machines (SVM) using MATLAB built-in functions *fitcsvm* and *fitcecoc*. To account for the fact that animals had different numbers of cells, we randomly selected 37 cells from each animal, given that this was the minimum number of cells detected in one animal, and report the mean results from 100 independent iterations. To balance the data, we used the same amount

of data for each CS and baseline period, each with 30 s windows binned in 1 s. Intraday decoders were validated using a ten-fold cross-validation procedure using MATLAB function *crossval*, in which decoders are trained on a partition of 90% of the data and tested on the remaining 10%, this procedure is carried out 10 times. Within each day, we trained a multiclass decoder to decode baseline vs. CS+ vs. CS− using different subsets of interneurons (e.g. only CS responsive or only non-CS responsive). As a control, we trained SVMs on temporally shuffled CS+, CS− and baseline labels. Additionally, to rule out the possibility that differences in non-CS decoder accuracy were due to fewer available cells, we trained control decoders on a random sample taken from all interneurons, matching the number of cells used in the non-CS decoders.

To compare how CS+ and CS− coding changed across days, we trained separately two-way decoders (baseline vs. CS+ and baseline vs. CS−) and used them to predict the stimulus identity during the other sessions. Overall accuracy across days was calculated as the sum of the diagonal of the confusion matrix (i.e. the number of true positives) divided by the overall sum of the confusion matrix. Using the confusion matrix, we also report precision (i.e. True positives/(True positives + False positives)), recall (i.e. True positives/(True positives + False negatives)) and F1 score (2 × (precision × recall)/(precision + recall)) for each class decoded.

Finally, we used decoder weights as a proxy to indicate the contribution of each interneuron to distinguishing between the trained classes (baseline, CS+ and CS−). Thus, to investigate interneuron selectivity for CS+ or CS−, we obtained the corresponding absolute decoding weights from the baseline vs. CS+ and baseline vs. CS− two-way decoders for each interneuron and calculated the correlation between them. A negative correlation between the weights would suggest stimulus selectivity[34]. We report the average correlation of each animal from 100 iterations. Furthermore, as a control, we calculated the correlation between CS+ decoding weights at the *ith* and *ith + 1* iteration of decoders trained on a randomly selected partition of 90% of the data from all interneurons. Average correlation between CS + decoding weight at the *ith* and *ith + 1* iterations were calculated per animal ($N = 9$ mice), using a total of 200 iterations to obtain 100 correlation values.

## Behaviour analysis

Pose estimation was performed using DeepLabCut version 2.2.2[73]. A ResNet-50-based neural network[74] was employed, with the network running for 1,060,000 training iterations. For training, locations of eight mouse body parts were manually labelled in video frames: nose, base of miniature microscope, left ear base, right ear base, neck, upper spine, middle spine, and tail base. A total of 146 frames were labelled from eight videos, which included both behavioural contexts and lighting conditions. Of these labelled frames, 95% were used for training. Post-processing of DeepLabCut output files, which included x/y coordinates and likelihood values for each body part, involved filtering low likelihood positions. These positions were replaced with the most recent highly probable positions. The displacement of each point over time was then calculated. Periods of pause were identified based on the displacement thresholds of five tracked points: left ear base, right ear base, upper spine, middle spine, and tail base. All points had to meet a threshold for displacement (0.7–3.5) to be considered in the analysis. Immobility was defined as a period during where the animal did not move its body for at least 2 s. The automated behavioural scoring was subsequently validated by a human scorer to ensure accuracy and reliability of the pose estimation results. A discrimination score DS was calculated based on CS+ and CS− immobility during Test/Extinction 1 (high fear state) as $DS = \frac{Imm_{CS+} - Imm_{CS-}}{Imm_{CS+} + Imm_{CS-}}$. Spearman correlation was used to analyse the relationship of neuronal responses with behavioural performance.

## Statistical analysis and data presentation

The number of analysed cells is indicated with '*n*', while '*N*' declares the number of animals. Averaging across multiple trials per cell/animal is indicated in the figure legends and respective methods sections where applicable. Reported *n/N* numbers always refer to data from individual cells/animals, no samples were measured repeatedly for statistical analysis. No statistical methods were used to predetermine sample sizes.

Statistical analysis was carried out using R (R 4.1.0, RStudio 2023.12.1), MATLAB (R2021b) or Prism 10 (GraphPad Software). All datasets were tested for Gaussian distribution using the Shapiro Wilk test. Equal variance across samples was assessed with Levene's test. Overall, the data did not pass normality or homoscedasticity tests, thus we report only non-parametric test results. For comparisons of multiple groups in a repeated measures variable, we used a Friedman test followed by Dunn's multiple comparisons. Comparisons of individual cellular responses to specific stimuli were analysed with paired Wilcoxon tests, comparisons between subpopulations with Mann–Whitney tests. In both cases, *p* values were adjusted with Bonferroni correction. To assess whether the distribution of cells in defined activity patterns was significantly different between stimuli or interneuron subpopulations, we applied Chi-Square tests. If significant, proportions were compared in a pairwise manner using the R function *prop.test*, which calculates a Chi-Square with Yates continuity correction for small expected *n* numbers ($n < 5$). All statistical tests were two-sided. Statistical significance threshold was set at 0.05 and significance levels are presented as *($p < 0.05$), **($p < 0.01$) or ***($p < 0.001$) in all figures. Statistical tests and results are reported in the respective figure legends, as well as in Supplementary Table 1.

Contrast and brightness of representative example images were minimally adjusted equally across the entire image using ImageJ. For figure display, confocal images were further scaled (0.5 × 0.5), and calcium activity was resampled to 4 Hz (traces and heatmaps). Averaged traces are displayed as mean with s.e.m. Violin plots illustrate distribution of all data points. Tukey box-and-whisker plots show median values, 25th and 75th percentiles, and min to max whiskers with exception of outliers (beyond 1.5 times interquartile range).

## Reporting summary

Further information on research design is available in the Nature Portfolio Reporting Summary linked to this article.

## Data availability

The data generated in this study has been deposited on Zenodo (https://doi.org/10.5281/zenodo.17390680). The source data underlying main figures and supplementary information are available as Source Data file. Source data are provided with this paper.

## Code availability

Custom-written code used for analysis and generating figures has been deposited on Zenodo (https://doi.org/10.5281/zenodo.17390682).

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

## Acknowledgements

The authors thank all members of the Krabbe lab for helpful discussions and comments. They thank Olga Sharma, Tobias Eichlisberger, Christian Müller and all staff of the FMI and DZNE Animal Facilities for excellent technical assistance. They further thank the Facility for Imaging and Microscopy at the FMI and the Light Microscopy Facility at the DZNE for their support with data acquisition and analysis, and Isaac Samuel Racine for advice on statistics. pAAV.CAG.Flex.GCaMP6f.WPRE.SV40 was a gift from Douglas Kim & GENIE Project (Addgene viral prep #100835-AAV9; http://n2t.net/addgene:100835; RRID:Addgene_100835). They are grateful to the GENIE Program at Janelia Research Campus of the Howard Hughes Medical Institute for making GCaMP6 available. The research was supported by the following funding agencies and institutions: Deutsches Zentrum für Neurodegenerative Erkrankungen (DZNE), Bonn (SKr, JG); Friedrich Miescher Institute for Biomedical Research (FMI), Basel (AL); Chan Zuckerberg Initiative, Ben Barres Early Career Acceleration Award 2023-331762 (SKr); Brain & Behavior Research Foundation, Young Investigator Award 31292 (SKr); Dementia Research Switzerland—Synapsis Foundation, Career Development Award 2020-CDA04 (SKr); Deutsche Forschungsgemeinschaft (DFG), SFB 1089 Teilprojekt (SKr, JG) and SPP 2411 Teilprojekt (JG); iBehave Network—sponsored by the Ministry of Culture and Science of the State of North Rhine-Westphalia (SK, JG); Swiss National Science Foundation (SNSF) Grant 310030_189123 and Adv Grant TMAG-3_209270 (AL); European Research Council, Starting Grant, AXPLAST (JG); The funders had no role in study design, data collection and analysis, decision to publish or preparation of the manuscript.

## Author contributions

Conceptualization: Y.B., J.G., A.L. and S.Kr.; Methodology: N.F., J.C.M., Y.B., J.G. and S.Kr.; Investigation: N.F., J.C.M., C.P., S.Ke., J.G. and S.Kr.; Analysis: N.F., J.C.M., C.P., S.Ke., B.E., Y.B., J.G. and S.Kr.; Visualisation: N.F., J.C.M. and S.Kr.; Funding acquisition: A.L., J.G. and S.Kr.; Supervision: S.Kr. and A.L.; Writing—original draft: N.F. and S.Kr.; Writing—review & editing: all authors.

## Funding

## Competing interests

The authors declare no competing interests.
