## [Transparent Peer Review file · Nature Communications]

Heterogeneous plasticity of amygdala interneurons in associative learning and extinction

Corresponding Author: Dr Sabine Krabbe

Version 0:

Reviewer comments:

Reviewer #1

(Remarks to the Author)

This study presents valuable insights into the heterogeneous plasticity of BLA interneurons during associative learning and extinction, utilizing deep-brain calcium imaging to monitor large populations of interneurons in freely moving mice. The authors identify both single-cell and population-level plasticity in BLA interneurons, as well as functional differences between distinct molecular subpopulations (VIP vs. SST) in relation to fear states.

Major Concerns:

1. The role of BLA interneurons in fear learning, particularly the involvement of PV, SOM, and VIP subtypes, has been well-explored in previous studies (e.g., Wolff et al., 2014; Krabbe et al., 2019). This study does not provide fundamentally novel insights regarding the importance of interneurons in fear processing. While the concept of heterogeneous plasticity in BLA interneurons is a noteworthy observation, it appears to be a generalizable phenomenon that could extend to a broad range of brain cell types observed through calcium imaging. Therefore, the novelty of the study lies primarily in demonstrating the phenomenon in BLA interneurons.
2. The authors utilize miniscope calcium imaging to monitor neuronal activity, a well-established technique for longitudinal studies of freely moving animals. While the population-level analysis presented is convincing, the manuscript does not address how neural plasticity at the single-cell level correlates with behavioral changes. To enhance the impact of the study, it would be beneficial to include a more detailed analysis of plasticity in the same, tracked cells across different stages of learning. Additionally, correlating these cellular plasticity changes with specific behavioral outcomes would provide a deeper understanding of the functional significance of the observed neural dynamics.

Minor Concerns:

1. Figure 1b: The histological analysis currently includes only a single neuron as an example. To strengthen the data presentation, the authors should provide a broader view of the immunohistochemical subpopulations. Additionally, statistical data quantifying the proportions of each subpopulation should be included to support the interpretation.
2. Figure 1b: This panel suggests that the authors examined all the cell types listed in Panel 1b, yet in the subsequent figures, only two of these cell types are imaged. Clarification is needed regarding why the other two cell types were excluded from the imaging experiments.
3. Line 94: Previous study (Shen et al. Nat. Med., 2019) reports that CCK+ neurons in the amygdala are predominantly glutamatergic (94.7%), which seems to contradict the conclusions drawn in this manuscript. The authors should revisit the histological data and provide clear statistical details to address this discrepancy.
4. Figure 1d: The term "resulting cell map" in Panel 1d requires further clarification. Specifically, how was this map generated. Providing additional detail in the Methods would aid in the interpretability of the figure.
5. Figure 1e: The legend for the arrowheads in Panel 1e is currently placed in Panel 1f, which could lead to confusion. Since readers typically examine Panel 1e before 1f, it would be clearer to relocate the legend to Panel 1e for better clarity and consistency.
6. Example Videos of Calcium Activity: Including example videos of calcium activity in response to CS and US would provide important insight into the dynamic changes in neural activity.
7. BLA PNs have been extensively studied, both at the population and single-cell level, and have been shown to exhibit heterogeneous plasticity as well. A discussion on how the findings in BLA INs compare with the plasticity observed in PNs would strengthen the manuscript. The authors should address these differences in plasticity and how they might contribute to the broader understanding of fear learning.

8. In the method regarding clustering analysis, the author used different number (16 for US, 8 for CS+ and CS-, 10 for both CS+ and CS-) for K-means clustering to categorize different response patterns. Could you elaborate on why these specific numbers of clusters were chosen for each group? Were they determined empirically, or was there a particular rationale or criteria for selecting these values?

Reviewer #2

(Remarks to the Author)

Reviewer #3

(Remarks to the Author)

This is a technically impressive study of GABAergic population activity in the BLA during fear and extinction learning in the mouse. Building on prior work, the authors obtain deep brain imaging from GAD expressing cells and two molecular subtypes. One of the strengths of the paper is the comprehensive evaluation of activity – previously what types of responses, CS vs. US, and in what proportions they exist, were not available or not characterized to this degree. The authors have delineated response patterns from among the GAD+ populations but also their dynamics across different sessions – e.g. some units have stable US activity, while others show experience-dependent adaptation. Some principles emerging from other high density recordings in other systems seem to be reflected. Namely, population turnover seems to (somehow) preserve stimulus encoding and can acquire stimulus responses following learning despite not receiving convergent CS-US information. And finally, one of the more interesting and novel result is that SST interneurons seem to exhibit greater activity following extinction (with a caveat below). However, reaching the end of this paper and considering all the points raised, I am finding it difficult to articulate exactly what important concepts have been learned – perhaps that response heterogeneity is independent of molecular subtype? While I regard this as an important concept, I don't think it is necessarily surprising and I am not sure how it will otherwise change experimental approaches to this system.

1. I am curious about the A-P spread of GRIN lenses. Perhaps more information could be included in suppl fig 1. The basal amygdala is itself functionally heterogeneous, containing a magnocellular division with abundant mPFC connections and a parvocellular division that does not. This might be relevant to components of the interneurons participating in fear expression and extinction.
2. The authors do a nice job separating stimulus-related activity into different buckets. Though the paper focuses on high and low fear states, however, they do not examine the relationship between interneuron activity and behavior (i.e. freezing). How do individual cells contribute to real-time behavioral events and is this pattern stable?
3. The most abundant interneurons (PV cells) were not analyzed.
4. Knowing the responses of SST and VIP cells across conditioning and extinction raises the question of functional relevance. Are these outcomes reflective of competition within local excitatory populations and are they causal to behavioral fear expression? The paper is limited in terms of what can be concluded except that these cell types are unique. The statistical quantification of these activity differences is also not clear. Differences between SST and VIP proportions were tested (fig 8d) but I don't see any analysis supporting changes within these populations between fear and extinction.
5. There was limited interpretation or speculation about the functional significance of different response types. This creates the impression that this information, at least at this stage, is not necessarily informative about memory-related processes and it is more of an open-ended question.

Reviewer #4

(Remarks to the Author)

This manuscript by Favila et al. investigates the role of interneurons in the amygdala during fear learning and extinction. The study employs a miniscope to track neuronal activity and reports mixed selectivity to different stimuli alongside complex plastic responses. The findings contribute to our understanding of how specific interneuron populations, namely VIP and SST, differentially respond to fear-related cues. In this paper, they conclude that VIP interneurons are predominantly activated in high-fear states, whereas SST interneurons show a preference for safety cues and suppress excitatory neuron responsiveness.

The authors analyze a large population of interneurons over several days, allowing for the study of neuronal dynamics beyond previous work. Interneuron responses appear highly dynamic, showing chance-level mixed selectivity, but after learning, their preference for CS+ over CS- remains stable. VIP interneurons demonstrate stronger activation to the US, with 50% of the population classified as “activated stable CS” cells, and their CS+ activation is subsequently suppressed after extinction. SST interneurons, in contrast, are more inhibited by the US, uniquely display an “activated up CS” category, and show stronger CS+ inhibition, which shifts to activation after extinction. The study suggests that VIP interneurons signal novel and fear-conditioned states, whereas SST interneurons signal safety during extinction.

The study presents novel findings with a robust methodology and well-executed controls, making it compelling and suitable for this journal; however, I have concerns regarding the terminology used and certain conclusions drawn from the data.

1. The term “mixed selectivity” should be clarified, as the study does not demonstrate that these neurons are more mixed-selective than a random population. In particular, since this effect is equal to chance, why do the authors show it as a main result?

2. The statement “CS+ evolved independently of US activation” is somewhat misleading, as CS+ dynamics are dependent on US activation at the population level (even if single CS+ cells can evolve without responding to the US). This distinction should be made clearer.

3. While percentages are reported for neuronal responses before and after conditioning, presenting raw numbers would be important to assess whether the sample size remains consistent and to understand the evolution of activation patterns. For example, the statement that “a larger fraction was selectively activated by CS- only before conditioning” could be misleading. The authors should show in parallel the raw numbers of activated neurons for CS+ and CS- before conditioning and compare them to show that these numbers are equal. If the baseline activation of CS- was higher than CS+, this could suggest an intrinsic difference between the tones rather than an experimental effect.

4. Figures 3 and 4: Could some of the observed differences in interneuron responses be driven by variability between animals rather than within animals? Reporting the percentage of neurons in each cluster per animal might help clarify this.

5. Figure 5: Would it be beneficial to correlate calcium responses of “high fear state” and “low fear state” neurons within individual mice or across different time points in extinction sessions? Why not also look at the response of neurons over extinction learning as well? This could strengthen the claim that these neurons encode fear states rather than other factors.

Minor points:

Line 124: The figure reference appears incorrect; should it be S1b?

Figure 2H: Would adding a correlation test help quantify whether US responses differ significantly from CS+ and CS-?

Reviewer #5

(Remarks to the Author)

Version 1:

Reviewer comments:

Reviewer #1

(Remarks to the Author)

I thank the authors for their thoughtful revisions. The new data and analyses significantly enhance the study's impact and conceptual depth, particularly the single-cell analyses linking neural activity to behavior.

While I am satisfied with the revisions, I maintain that the core finding on heterogeneous BLA interneuron plasticity, though well-executed, builds directly on concepts from prior acute and population-level studies.

Furthermore, the authors have not addressed the major concern raised by Reviewer #3 regarding the novel finding mentioned at the end of the first paragraph.

(Remarks on code availability)

Reviewer #2

(Remarks to the Author)

(Remarks on code availability)

Reviewer #3

(Remarks to the Author)

The authors have addressed my concerns and the new data provided help strengthen the impact.

(Remarks on code availability)

Reviewer #4

(Remarks to the Author)

The authors have satisfactorily addressed all previous concerns, and the manuscript has improved significantly. I recommend publication in its current form.

(Remarks on code availability)

Reviewer #5

(Remarks to the Author)

(Remarks on code availability)

/

Rebuttal Letter

Heterogeneous plasticity of amygdala interneurons in associative learning and extinction

Structure: Reviewer comments: *black*; our replies: blue

General note

We would like to thank all reviewers for their positive and constructive feedback. The reviewers raised many interesting points and we are confident that we addressed all of these issues. We have revised the manuscript according to their suggestions and provide new data and analyses that further confirm and extend our original conclusions. Some of our new analyses and experiments are:

1. By investigating interneuron activity during spontaneous immobility bouts, we explored whether BLA interneurons responsive to conditioned auditory cues exhibit distinct activity reflecting internal fear states. Notably, interneurons suppressed by fear cues or activated during extinction are also modulated by internal behavioural state, suggesting a role in transitions out of fear or in safety signalling. In contrast, CS+ 'Fear activated' and 'Extinction inhibited' interneurons are less sensitive to internal state transitions, consistent with a role in representing cue-specific associative information rather than general fear levels.
2. Using this analysis, we additionally show that VIP interneurons are selectively activated to external stimuli, but not modulated by internal fear states, suggesting that VIP interneurons are mediating cue-driven disinhibition to salient stimuli. In contrast, SST interneuron activity additionally aligns closely with internal behavioural states, indicating that they can suppress excitatory neuron responses in safety conditions. We have revised our manuscript accordingly to highlight these distinctive functions of SST and VIP interneurons.
3. We report that the size of specific interneuron plasticity clusters correlates with behavioural outcomes, in particular with the ability to discriminate between predictive CS+ and control CS- cues.
4. We have added a detailed analysis addressing plasticity of interneurons during extinction training (GAD2, SST, VIP).
5. We are providing new histological data quantifying marker expression in the GCaMP+ population targeted with the GAD2-Cre mouse line.

Below, we address all of the reviewer's comments point by point. Changes to the manuscript and figure legends are highlighted in red in the respective files.

Reviewer #1 (Remarks to the Author):

This study presents valuable insights into the heterogeneous plasticity of BLA interneurons during associative learning and extinction, utilizing deep-brain calcium imaging to monitor large populations of interneurons in freely moving mice. The authors identify both single-cell and population-level plasticity in BLA interneurons, as well as functional differences between distinct molecular subpopulations (VIP vs. SST) in relation to fear states.

Major Concerns:

1. The role of BLA interneurons in fear learning, particularly the involvement of PV, SOM, and VIP subtypes, has been well-explored in previous studies (e.g., Wolff et al., 2014; Krabbe et al., 2019). This study does not provide fundamentally novel insights regarding the importance of interneurons in fear processing. While the concept of heterogeneous plasticity in BLA interneurons is a noteworthy observation, it appears to be a generalizable phenomenon that could extend to a broad range of brain cell types observed through calcium imaging. Therefore, the novelty of the study lies primarily in demonstrating the phenomenon in BLA interneurons.

We respectfully disagree with the reviewer's statement and the citation of the literature: So far, no study has addressed across-day interneuron plasticity in multi-day fear learning and extinction. Wolff et al. 2014 or Krabbe et al. 2019 investigated neuronal activity acutely during the fear conditioning day, and only on the population level. The present work is novel and the first in-depth study showing that individual amygdala interneurons exhibit heterogeneous and complex plastic responses during the acquisition, expression and extinction of aversive memories.

In addition, to go beyond classical CS-response plasticity, we extended our analysis to the level of behavioural encoding by BLA interneurons. We now provide new data showing that interneurons that are inhibited by fear-associated cues or recruited during extinction also track internal behavioural states, suggesting that they play a role in signalling transitions between fear and safety, while fear activated neurons are selectively modulated by cue-specific associative information. This extends our knowledge of the encoding of fear and safety states to amygdala interneurons.

The reviewer makes an important point, i.e., that heterogeneous functional plasticity in defined cell types is most likely a generalisable phenomenon. However, for many (inhibitory) cell types and brain areas this is not known (or appreciated) and a timely field of study. This is exactly why our findings are novel and impactful, given the importance to understand how inhibitory plasticity mechanisms contribute to learning across brain regions.

The novelty of this study further lies in demonstrating, for the first time, that such dynamic plasticity occurs in BLA interneurons in a genetic subtype-specific manner during fear learning – providing important insight into circuit-level mechanisms of learning that likely extend beyond the amygdala. As detailed below, we are now providing new analyses that emphasise functional differences between BLA interneuron subtypes, demonstrating that VIP interneurons are selectively tuned to salient external sensory cues, particularly during the acquisition and expression of aversive associations, while SST interneurons additionally track internal behavioural states and contribute to the maintenance of safety-related inhibition during extinction.

2. The authors utilize miniscope calcium imaging to monitor neuronal activity, a well-established technique for longitudinal studies of freely moving animals. While the population-level analysis presented is convincing, the manuscript does not address how neural plasticity at the single-cell level correlates with behavioral changes. To enhance the impact of the study, it would be beneficial to include a more detailed analysis of plasticity in the same, tracked cells across different stages of learning. Additionally, correlating these cellular plasticity changes with specific behavioral outcomes would provide a deeper understanding of the functional significance of the observed neural dynamics.

We thank the reviewer for these suggestions. To briefly clarify, all neurons included in the analysis of this manuscript (i.e., single cell as well as population level analysis) were tracked across all stages of learning.

To address correlations with behavioural outcomes, we added the following analyses:

1. To explore whether conditioned cue-responsive BLA interneurons reflect internal fear states, we analysed their calcium dynamics during spontaneous immobility bouts outside of CS periods (Rebuttal Figure 1.1). We find that CS+ plastic BLA interneurons, particularly the 'Fear inhibited' and 'Extinction activated' clusters, are modulated not only by conditioned cues but also by internal fear states inferred by spontaneous immobility suggesting that these neurons integrate sensory associations with internal state dynamics. On the single neuron level, this activity pattern is further stable across experimental days. This state-dependent activity suggests that they gate fear expression by modulating PN excitability rather than encoding associative information *per se*. In contrast, CS+ 'Fear activated' and 'Extinction inhibited' neurons are not changing their activity during immobility bouts and appear less sensitive to internal state transitions, pointing to a role in representing cue-specific associative information without directly influencing the balance between generalisation and discrimination. This data is shown in new Figure 6 and new Supplementary Figure S7e-f.

Rebuttal Figure 1.1: Encoding of fear states in amygdala interneurons. **a**, Average traces (top) and heatmap of basolateral amygdala interneuron activity during immobility bouts, averaged across all events in Test/Extinction 1 and Extinction 2 sessions, showing neuronal responses aligned to immobility start and movement onset (= immobility stop). Neurons are sorted by response amplitude upon immobility start ($n = 519$ cells from $N = 9$ mice). **b**, Average traces sub-selected for across-day CS+ plasticity clusters during immobility (see Figure 5; 'Fear/activated after conditioning, $n = 66$ '; 'Fear inhibited'/inhibited after conditioning, $n = 48$ '; 'Extinction'/activated after extinction, $n = 63$ '; 'Extinction inhibited'/inhibited after extinction, $n = 33$). **c**, Corresponding area under the curve (AUC) for immobility start and stop (2 s duration). Start: Kruskal Wallis test ($H = 10.83$), $p = 0.0127$ with *post hoc* comparison (Fear vs. Fear inhibited, $p = 0.0173$). Stop: Kruskal Wallis test ($H = 19.86$), $p = 0.0002$ with *post hoc* comparison (Fear vs. Fear inhibited, $p = 0.0015$; Fear inhibited vs. Extinction inhibited, $p = 0.0015$). **d**, Example correlation of individual neuronal responses to immobility start (left) and stop (right) between Test/Extinction session 1 (Ext 1) and Extinction 2 (Ext 2), lines indicate linear regression fit (shown for visualisation); correlation coefficient ρ reflects Spearman's rank-order correlation ($n = 93$ cells from mouse ID 710407). **e**, Average correlation matrix of single cell responses to immobility start and stop across different stages of conditioning, dots indicate a significant difference of the correlation from 0 across $N = 9$ animals (Hab, habituation; Ext, extinction). Average traces in **a** and **b** are mean with s.e.m.; violin plots in **c** show distribution of all data points, Tukey box-and-whisker plots median values, 25th and 75th percentiles, and min to max whiskers with exception of outliers, dots indicate the mean. * $p < 0.05$, ** $p < 0.01$.

2. To investigate whether differences in molecular interneuron subtype responses extend beyond stimulus-driven activation, we examined SST and VIP interneuron activity during spontaneous immobility outside of CS cue periods (Rebuttal Figure 1.2). SST interneurons

were suppressed during immobility and reactivated with movement onset, indicating an additional role in internal state transitions and safety signalling, while VIP interneurons were largely unmodulated, supporting their specialisation for externally driven disinhibition to aversive and predictive cues. These findings highlight a dynamic, state-dependent engagement of the canonical VIP→SST→PN circuit motif, which is likely shaped by distinct input pathways to promote functional biases of each subtype to flexibly modulate BLA output and plasticity during fear and extinction. This data is presented in new Figure 9f-h.

Rebuttal Figure 1.2: Encoding of fear states in molecular interneuron subpopulations. **a**, Average traces (top) and heatmap of SST interneuron activity during immobility bouts, averaged across all events in Test/Extinction 1 and Extinction 2 sessions, showing neuronal responses aligned to immobility start and movement onset (= immobility stop). Neurons are sorted by response amplitude upon immobility start ($n = 114$). **b**, Same for VIP interneuron activity ($n = 152$). **c**, Corresponding area under the curve (AUC) for immobility start and stop (2 s duration). Kruskal Wallis test ($H = 15.33$), $p = 0.0016$ with *post hoc* comparison, SST start vs. SST stop, $p = 0.0025$; SST stop vs. VIP stop, $p = 0.0177$. Average traces **a** and **b** are mean with s.e.m.; violin plots in **c** show distribution of all data points, Tukey box-and-whisker plots show median values, 25th and 75th percentiles, and min to max whiskers with exception of outliers, dots indicate the mean. * $p < 0.05$, ** $p < 0.01$.

3. We examined the relationship between interneuron recruitment and behavioural performance of individual mice (Rebuttal Figure 1.3), and found that animals with a higher fraction of CS+ 'Fear inhibited' neurons exhibited significantly better discrimination between CS+ and CS-, which is consistent with the idea that transient removal of tonic inhibition during the expression of fear supports the accurate encoding of threats. Conversely, mice with more 'Extinction activated' neurons tended to perform more poorly. Strikingly, this reflects a pre-existing inhibitory bias that limits fear expression in animals with less precise associative memory. These findings highlight a potential inhibitory circuit-level substrate for variability in fear discrimination and generalisation between individuals. This data is shown in new Supplementary Figure S7a-d.

Rebuttal Figure 1.3: Relationship between interneuron plasticity and behavioural performance. **a-b**, Correlation of fractions of neurons in across-day CS+ plasticity clusters with learning-related performance in individual animals ($N = 9$) for **a**, 'Fear' inhibited interneurons and **b**, 'Extinction' interneurons. Lines indicate linear regression fit (shown for visualisation); correlation coefficient ρ reflects Spearman's rank-order correlation.

4. We now provide an extended analysis of plasticity in individual neurons, addressing how neuronal activity changes during extinction learning. This shows that the responses of 'Fear' and 'Fear Inhibited' clusters weakened during initial extinction, but partially recovered after 24 hours, mirroring spontaneous recovery at the behavioural level (Rebuttal Figure 1.4a). By contrast, extinction-related responses emerged more strongly by the end of the second extinction session, indicating a shift in network dynamics. We further find distinct functional

subgroups, including a separate group of neurons which were activated predominantly in extinction session 1, while extinction inhibited cells were grouped in a cluster inhibited across all CS+ presentations in extinction session 2 (Rebuttal Figure 1.4b-c). Notably, 'Extinction 1 activated' neuron activity was selectively enriched in VIP interneurons (Rebuttal Figure 1.4d), stressing their importance for signalling fear-related external cues. These new results highlight a dynamic, experience-dependent reorganisation of interneuron activity during fear extinction learning. This data is shown in new Figure 5j, new Supplementary Figure S6g-h and new Supplementary Figure S13.

Rebuttal Figure 1.4: Plasticity of amygdala interneurons in extinction learning. **a**, Development of interneuron responses across extinction sessions in *GAD2-Cre* mice. Averages for four CS+ presentations are shown as indicated. All cells, $n = 519$; Fear, $n = 66$; Fear inhibited, $n = 48$; Extinction, $n = 63$; Extinction inhibited, $n = 33$. **b**, Heatmap of CS+ responses in BLA interneurons from in *GAD2-Cre* mice clustered into groups depending on their response pattern across the two extinction sessions ($n = 519$; Extinction 1 activated, $n = 93$; Fear activated, $n = 116$; Fear inhibited, $n = 109$; Extinction activated, $n = 81$; Extinction inhibited, $n = 120$). **c**, Corresponding proportion of interneurons in CS+ extinction clusters ($n = 519$). **d**, Proportion of cells in CS+ extinction clusters for SST and VIP interneurons (SST, $n = 114$; VIP, $n = 152$) based on recordings from *SST-Cre* and *VIP-Cre* mice. Chi-Square test ($\chi^2(4) = 19.773$), $p = 0.0006$; SST vs. VIP, *post hoc* Chi-Square test with Bonferroni correction, Extinction 1 activated, $p = 0.0028$. Average traces are mean with s.e.m.; ** $p < 0.01$, *** $p < 0.001$.

Minor Concerns:

1. Figure 1b: The histological analysis currently includes only a single neuron as an example. To strengthen the data presentation, the authors should provide a broader view of the immunohistochemical subpopulations. Additionally, statistical data quantifying the proportions of each subpopulation should be included to support the interpretation.

We have now included a detailed histological analysis and determined how many GCaMP+ neurons in *GAD2-Cre* mice (after AAV2/9.CAG.flex.GCaMP6 injection) were immunopositive for PV, SST, CCK and VIP. This data is shown in the new Supplementary Figure S1 (Rebuttal Figure 1.5). On average, 26.0% of GCaMP+ neurons were PV+, 15.3% SST+, 12.0% CCK+ and 16.4% VIP+ (data from $N = 3$ mice). This matches previously reported fractions of PV, SST, CCK and VIP interneurons in the general BLA inhibitory population (Hájos, 2021). The remaining fraction of GCaMP+ interneurons without co-expression of these selected marker genes likely represent other subpopulations negative for these markers, such as neurogliaform-like cells, *Npy*+ interneurons, *Scng*+ interneurons or calretinin interneurons negative for VIP and CCK (Hochgerner et al., 2023; Perumal and Sah, 2021).

Rebuttal Figure 1.5: Expression of GCaMP6 in *GAD2-Cre* mice. **a**, Representative example of GCaMP6 expression in the amygdala after injection of AAV2/9.CAG.flex.GCaMP6f in *GAD2-Cre* mice. Maximum intensity projection; outline delineates the BLA. **b**, Examples of GCaMP6 co-expression with main BLA interneuron marker genes from the same mouse (VIP, vasoactive intestinal peptide; CCK, cholecystokinin; SST, somatostatin; PV, parvalbumin). Arrowheads point to GCaMP6+ neurons immunopositive for the respective marker. Note that the peptide CCK is not exclusively expressed in interneurons but also in excitatory BLA projection neurons (Reeb et al., 2023). **c**, Fraction of GCaMP6+ neurons immunopositive for VIP, CCK, SST and PV in the BLA after AAV injection in *GAD2-Cre* mice. PV and SST expression were analysed in the same sections (N = 3 mice, n = 541 \pm 53 GCaMP6+ cells), and CCK and VIP together in adjacent sections of the same mice (n = 624 \pm 97 cells). Dots are individual animals; line indicates the mean. **d**, Combined fraction of GCaMP6+ neurons immunopositive for VIP, CCK, SST or PV (mean of N = 3 mice).

2. *Figure 1b: This panel suggests that the authors examined all the cell types listed in Panel 1b, yet in the subsequent figures, only two of these cell types are imaged. Clarification is needed regarding why the other two cell types were excluded from the imaging experiments.*

This might be a misunderstanding. We did not exclude cell types from imaging. With the *GAD2-Cre* mouse line, we are targeting all interneuron types in the BLA irrespective of their molecular marker genes (data presented in Main Figures 1-7). The example shown in Figure 1 illustrates that all major BLA interneuron subtypes are included in our analysis with this approach, with fractions comparable to the general inhibitory population (see above).

For a detailed analysis of response patterns in defined molecular interneuron subtypes (data presented in Main Figures 8 & 9), we specifically selected SST and VIP interneurons, which form a canonical circuit motif across cortical areas that plays a central role in learning and experience-dependent plasticity (Kepecs and Fishell, 2014; Kullander and Topolnik, 2021). Within this circuit, VIP interneurons inhibit tonically active SST cells, which target the distal dendrites of PNs (Wolff et al. 2014). By modulating synaptic integration and plasticity at input sites in cortex and BLA (Karnani et al., 2016a, 2016b; Krabbe et al., 2019; Pi et al., 2013; Wolff et al., 2014), this disinhibitory architecture enables flexible, state-dependent control principal neuron activity and synaptic plasticity, which is essential for top-down control, sensory discrimination and learning (Adler et al., 2019; Artinian and Lacaille, 2018; Bastos et al., 2023; Chevy et al., 2024; Krabbe et al., 2019; Piet et al., 2024). Unlike PV and CCK interneurons, which predominantly mediate perisomatic inhibition and regulate spike timing, SST cells act at the level of dendritic inputs – a key site for plasticity induction during CS processing (d’Aquin et al., 2022; Cichon and Gan, 2015; Gentet et al., 2012; Wolff et al., 2014). Notably, SST and VIP interneurons in the BLA are reciprocally connected (Krabbe et al., 2019), and this mutual inhibition may facilitate switching between high- and low-fear states. We are now putting

stronger emphasis on our choice to specifically investigate how SST and VIP cells match to functional subtypes in the general GAD2+ inhibitory population in our revised manuscript.

3. *Line 94: Previous study (Shen et al. Nat. Med., 2019) reports that CCK+ neurons in the amygdala are predominantly glutamatergic (94.7%), which seems to contradict the conclusions drawn in this manuscript. The authors should revisit the histological data and provide clear statistical details to address this discrepancy.*

The reviewer is correct that most CCK+ neurons in the BLA are excitatory projection neurons, as demonstrated in numerous studies. However, we would like to clarify that our approach utilises the well-established GAD2-Cre mouse line, directing expression of Cre-dependent GCaMP only to interneurons (see also Taniguchi et al. 2011) – not a CCK-Cre mouse line which would target GCaMP expression to excitatory and inhibitory CCK+ neurons. Given that with this approach, Cre expression is restricted to neurons with Gad2 expression, only GABAergic inhibitory BLA neurons were targeted in our experiments. This is also visible in our new Supplementary Figure S1b (see Rebuttal Figure 1.5b), illustrating that only a small fraction of CCK+ cells are GCaMP+ (and thus interneurons).

4. *Figure 1d: The term “resulting cell map” in Panel 1d requires further clarification. Specifically, how was this map generated. Providing additional detail in the Methods would aid in the interpretability of the figure.*

“Resulting cell map” refers to all cell ROIs detected using Principal Component Analysis (PCA) and Independent Component Analysis (ICA) for a given animal. We have added a statement to the Methods section to make this clearer: “Cell detection was carried out using Principal Component Analysis (PCA) and Independent Component Analysis (ICA) with the CIAtah package. ROIs were initially oversampled and then manually inspected. ROIs not matching individual neurons in the motion-corrected video were removed. After confirming the individual cell ROIs, a cell map was generated by aggregating the spatial footprints of all identified cells within the field of view.” (line 796ff).

5. *Figure 1e: The legend for the arrowheads in Panel 1e is currently placed in Panel 1f, which could lead to confusion. Since readers typically examine Panel 1e before 1f, it would be clearer to relocate the legend to Panel 1e for better clarity and consistency.*

We have changed the figure legend according to the reviewer’s suggestion.

6. *Example Videos of Calcium Activity: Including example videos of calcium activity in response to CS and US would provide important insight into the dynamic changes in neural activity.*

We have uploaded an example video as suggested (new Supplementary Movie 1). This shows the first and last CS+/US pairings during fear conditioning of the example mouse depicted in Figures 1 and 2.

7. *BLA PNs have been extensively studied, both at the population and single-cell level, and have been shown to exhibit heterogeneous plasticity as well. A discussion on how the findings in BLA INs compare with the plasticity observed in PNs would strengthen the manuscript. The authors should address these differences in plasticity and how they might contribute to the broader understanding of fear learning.*

We thank the reviewer for this suggestion and have extended our discussion related to comparison of PN and IN plasticity in fear and extinction learning (line 468ff), including a functional interpretation of different activity patterns.

8. In the method regarding clustering analysis, the author used different number (16 for US, 8 for CS+ and CS-, 10 for both CS+ and CS-) for K-means clustering to categorize different response patterns. Could you elaborate on why these specific numbers of clusters were chosen for each group? Were they determined empirically, or was there a particular rationale or criteria for selecting these values?

To identify subclasses of responses, we performed PCA with K-means clustering on the baselined traces of significantly responsive interneurons (using an explained variance of at least 80%). To evaluate cluster segmentation, we calculated the Silhouette coefficient for each cell, a metric that considers both intra-cluster cohesion and inter-cluster separation. We evaluated the number of clusters ranging from 2 to 20 using the MATLAB built-in function 'evalclusters'. Across stimuli, two clusters were identified as the optimal division based on the mean Silhouette coefficient. This cluster segmentation clearly divided the data into activated and inhibited responses (Rebuttal Figure 1.6a-d). However, we noticed that our data exhibited more heterogeneity than could be explained by this activated/inhibited classification alone. Therefore, we manually increased the number of clusters to better represent the diversity of response patterns, while maintaining a mean Silhouette coefficient close to that obtained with two clusters (Rebuttal Figure 1.6e-g).

Rebuttal Figure 1.6: Silhouette analysis of automatic and semi-automatic clustering of single cell CS+ responses. a, Average Silhouette values for K-means clustering across a range of cluster numbers ($k = 2-20$), showing the optimal automatic cluster number based on highest average Silhouette value is two clusters. b, Silhouette values of individual cells within each of the two automatic clusters. c, Average Silhouette values for each cluster at the optimal number ($k = 2$). d, Heat map of single cell CS+ responses during fear conditioning day grouped by the two automatically determined clusters, showing a simple division into activated and inhibited responses. e, Silhouette values of individual cells grouped into six curated clusters, obtained by merging automatic clusters to capture the heterogeneity of CS+ responses. f, Average Silhouette values of the final six curated clusters. For comparison, the red dotted line indicates the best average Silhouette value obtained from optimal automatic clustering (from panel a). g, Heat map of single cell CS+ responses grouped into the six clusters, highlighting the diversity of more response patterns.

Reviewer #2 (Remarks to the Author):

Reviewer #3 (Remarks to the Author):

This is a technically impressive study of GABAergic population activity in the BLA during fear and extinction learning in the mouse. Building on prior work, the authors obtain deep brain imaging from GAD expressing cells and two molecular subtypes. One of the strengths of the paper is the comprehensive evaluation of activity – previously what types of responses, CS vs. US, and in what proportions they exist, were not available or not characterized to this degree. The authors have delineated response patterns from among the GAD+ populations but also their dynamics across different sessions – e.g. some units have stable US activity, while others show experience-dependent adaptation. Some principles emerging from other high density recordings in other systems seem to be reflected. Namely, population turnover seems to (somehow) preserve stimulus encoding and can acquire stimulus responses following learning despite not receiving convergent CS-US information. And finally, one of the more interesting and novel result is that SST interneurons seem to exhibit greater activity following extinction (with a caveat below). However, reaching the end of this paper and considering all the points raised, I am finding it difficult to articulate exactly what important concepts have been learned – perhaps that response heterogeneity is independent of molecular subtype? While I regard this as an important concept, I don't think it is necessarily surprising and I am not sure how it will otherwise change experimental approaches to this system.

1. I am curious about the A-P spread of GRIN lenses. Perhaps more information could be included in suppl fig 1. The basal amygdala is itself functionally heterogeneous, containing a magnocellular division with abundant mpfc connections and a parvocellular division that does not. This might be relevant to components of the interneurons participating in fear expression and extinction.

The reviewer raises an interesting point. We now provide horizontal maps of our implant sites as well. The new Supplementary Figure S2c (Rebuttal Figure 3.1a) shows implant sites according to their exact dorso-ventral position. In new Supplementary Figure S6c (Rebuttal Figure 3.1b), we have spatially mapped interneuron types according to their across-day plasticity (tone, fear and extinction modulated neurons). Unlike excitatory projection neurons (Kim et al., 2016; Manoocheri and Carter, 2022; Zhang et al., 2020), we do not observe a preferential location of fear- and extinction-modulated interneurons in anterior or posterior BLA regions.

Rebuttal Figure 3.1: Horizontal mapping of lens implant sites. a, Schematic illustrating all reconstructed implant sites of GRIN lenses (lens front) on horizontal mouse brain atlas planes (N = 9 mice). LA, lateral amygdala; BA, basal amygdala; CEA, central amygdala; LV, lateral ventricle. b, Location of interneurons (centroids) on the horizontal atlas plane colour-coded for CS+ response type across days (n = 519 cells from N = 9 mice). For easier assessment, different ventral planes of lens fronts (see a) were combined on the same plane. Lines indicate BLA borders.

2. The authors do a nice job separating stimulus-related activity into different buckets. Though the paper focuses on high and low fear states, however, they do not examine the relationship between interneuron activity and behavior (i.e freezing). How do individual cells contribute to real-time behavioral events and is this pattern stable?

We now analysed the encoding of fear states in interneuron subpopulations. To this end, we have taken immobility epochs outside of any CS presentation to avoid confounding effects of sensory stimulation, and aligned neuronal responses to these real-time behavioural events. Our new Figure 6a shows that subpopulations of GAD2+ interneurons are opposingly active during immobility, with individual cells being inhibited by fear states and activated upon movement onset, and *vice versa* (Rebuttal Figure 3.2a). Subclusters of interneurons encoding sensory-induced fear states during CS+ presentation show matching responses during spontaneous immobility bouts: as illustrated in new Figure 6b-c, CS+ 'Fear inhibited' and 'Extinction activated' clusters are suppressed during immobility and activated when transitioning towards an exploratory/low-fear state with movement onset (Rebuttal Figure 3.2b-c). This is stable at the single cell level across habituation (albeit only few immobility events are present before conditioning), test/extinction session 1 and extinction session 2, as shown in new Figure 6d-e (Rebuttal Figure 3.2d-e). Furthermore, across-day CS+ plasticity clusters show the same activity pattern across test/extinction session 1 and extinction session 2 (see new Supplementary Figure S7e-h).

Rebuttal Figure 3.2: Encoding of fear states in amygdala interneurons. **a**, Average traces (top) and heatmap of basolateral amygdala interneuron activity during immobility bouts, averaged across all events in Test/Extinction 1 and Extinction 2 sessions, showing neuronal responses aligned to immobility start and movement onset (= immobility stop). Neurons are sorted by response amplitude upon immobility start ($n = 519$ cells from $N = 9$ mice). **b**, Average traces sub-selected for across-day CS+ plasticity clusters during immobility (see Figure 5; 'Fear'/activated after conditioning, $n = 66$; 'Fear inhibited'/inhibited after conditioning, $n = 48$; 'Extinction'/activated after extinction, $n = 63$; 'Extinction inhibited'/inhibited after extinction, $n = 33$). **c**, Corresponding area under the curve (AUC) for immobility start and stop (2 s duration). Start: Kruskal Wallis test ($H = 10.83$), $p = 0.0127$ with *post hoc* comparison (Fear vs. Fear inhibited, $p = 0.0173$). Stop: Kruskal Wallis test ($H = 19.86$), $p = 0.0002$ with *post hoc* comparison (Fear vs. Fear inhibited, $p = 0.0015$; Fear inhibited vs. Extinction inhibited, $p = 0.0015$). **d**, Example correlation of individual neuronal responses to immobility start (left) and stop (right) between Test/Extinction session 1 (Ext 1) and Extinction 2 (Ext 2), lines indicate linear regression fit (shown for visualisation); correlation coefficient ρ reflects Spearman's rank-order correlation ($n = 93$ cells from mouse ID 710407). **e**, Average correlation matrix of single cell responses to immobility start and stop across different stages of conditioning, dots indicate a significant difference of the correlation from 0 across $N = 9$ animals (Hab, habituation; Ext, extinction). Average traces in a and b are mean with s.e.m.; violin plots in c show distribution of all data points, Tukey box-and-whisker plots median values, 25th and 75th percentiles, and min to max whiskers with exception of outliers, dots indicate the mean. * $p < 0.05$, ** $p < 0.01$.

These results suggest that BLA interneurons which are suppressed by auditory cues during fear or activated by the CS+ during extinction are also modulated by the behavioural state of the animal. Their inhibition during spontaneous immobility and reactivation upon immobility offset suggests that they play a role in encoding transitions out of fear- or safety-related processes. By contrast, 'Fear activated' and 'Extinction inhibited' CS+ neurons appear less sensitive to internal state transitions, which points to a role in representing cue-specific associative information rather than general fear levels.

Analogous, we also analysed the responses of SST and VIP interneurons during spontaneous immobility epochs (new Figure 9f-h; Rebuttal Figure 3.3). In line with our result that SST interneurons are preferably activated in safety states, this interneuron subtype is predominantly inhibited during immobility and activated with movement onset upon transition to a low fear state, supporting our conclusion of state-encoding in SST interneurons. In contrast, we could not detect such modulation of VIP interneurons during immobility events. This leads us to the conclusion that VIP cells are more prone to respond to salient external cues, contributing to SST suppression for dendritic disinhibition of PNs in associative learning, instead of signalling internal high fear states. In contrast, SST interneurons additionally track internal behavioural states and are also suppressed in high fear during immobility, which might indicate a multiplexed role for sensory- and state-dependent gating of PN plasticity. These results suggest that the canonical VIP→SST→PN disinhibitory motif functions in a dynamic, state-dependent manner in the BLA, presumably shaped by distinct long-range inputs, allowing for state-dependent gating of PN plasticity. We have revised our manuscript accordingly to highlight these distinctive functions of SST and VIP interneurons, and a possible state-dependent role of diverse BLA microcircuits.

Rebuttal Figure 3.3: Encoding of fear states in molecular interneuron subpopulations. **a**, Average traces (top) and heatmap of SST interneuron activity during immobility bouts, averaged across all events in Test/Extinction 1 and Extinction 2 sessions, showing neuronal responses aligned to immobility start and movement onset (= immobility stop). Neurons are sorted by response amplitude upon immobility start ($n = 114$). **b**, Same for VIP interneuron activity ($n = 152$). **c**, Corresponding area under the curve (AUC) for immobility start and stop (2 s duration). Kruskal Wallis test ($H = 15.33$), $p = 0.0016$ with *post hoc* comparison, SST start vs. SST stop, $p = 0.0025$; SST stop vs. VIP stop, $p = 0.0177$. Average traces a and b are mean with s.e.m.; violin plots in c show distribution of all data points, Tukey box-and-whisker plots show median values, 25th and 75th percentiles, and min to max whiskers with exception of outliers, dots indicate the mean. $**p < 0.01$.

3. The most abundant interneurons (PV cells) were not analyzed.

Our study focuses on SST and VIP interneurons due to their central role in a well-defined, canonical disinhibitory microcircuit motif (Kepecs and Fishell, 2014; Kullander and Topolnik, 2021), which is crucial for learning and experience-dependent plasticity. VIP interneurons can inhibit tonically active SST cells which target the dendrites of excitatory PNs, thereby regulating synaptic integration and plasticity at distal input sites of PNs in cortex and the BLA (Karnani et al., 2016b, 2016a; Krabbe et al., 2019; Pi et al., 2013; Wolff et al., 2014). This disinhibitory motif enables the dynamic, state-dependent modulation of excitatory activity and plasticity, which is essential for tasks involving top-down modulation, sensory discrimination, and

learning (see for example Adler et al., 2019; Artinian and Lacaille, 2018; Bastos et al., 2023; Chevy et al., 2024; Krabbe et al., 2019; Piet et al., 2024). Unlike PV interneurons, which primarily mediate perisomatic inhibition and control spike timing, SST cells exert their influence at the level of dendritic inputs – a key locus for plasticity induction upon auditory stimulation (d'Aquin et al., 2022; Cichon and Gan, 2015; Gentet et al., 2012; Wolff et al., 2014). Given that BLA SST and VIP interneurons are reciprocally connected (Krabbe et al., 2019), mutual inhibition might additionally be an important mechanism supporting switches between high and low fear states – a concept we have now elaborated on in the manuscript as well. We fully agree that analysis of additional interneuron subtypes targeting the perisomatic region of PNs, such as CCK and PV interneurons, would be interesting, but we feel that this is beyond the scope of the present manuscript which focusses on disinhibitory dynamics and dendritic modulation during auditory associative learning mediated by the VIP→SST→PN disinhibitory circuit motif.

4. Knowing the responses of SST and VIP cells across conditioning and extinction raises the question of functional relevance. Are these outcomes reflective of competition within local excitatory populations and are they causal to behavioral fear expression? The paper is limited in terms of what can be concluded except that these cell types are unique. The statistical quantification of these activity differences is also not clear. Differences between SST and VIP proportions were tested (fig 8d) but I don't see any analysis supporting changes within these populations between fear and extinction.

The reviewer asks to demonstrate causality. An obvious experimental approach would be the use of optogenetic activation or inhibition of specific interneuron types during behaviour. However, we demonstrate that molecular interneuron subtypes such as SST and VIP are functional heterogeneous even within a given class – a general optogenetic approach targeting all cells of a given molecular subtype would uniformly drive the entire population and would artificially overrule their physiologically diverse response patterns. We therefore do not consider blanket optogenetics suitable to address this question. Accordingly, an approach that specifically targets individual cells of a given functional subtype within a genetic class during behaviour would be necessary. While all-optical approaches through GRIN lenses might be a solution to address this problem (Zhang et al., 2023), we would need to know which interneuron belongs to which functional cluster already before optical interrogation (e.g., which neuron is a 'Fear' interneuron). However, this cannot be known in advance since clustering BLA interneurons into functional subtypes depends on the full paradigm and includes neuronal responses before conditioning as well as after conditioning and after extinction. Therefore, an all-optical approach would at this point not be suitable to address this question either (also considering additional technical challenges in freely moving animals). Ideally, as indicated in the discussion, we would have additional marker genes for functional subpopulations that would allow to target these functional cell types with intersectional genetic approaches. Given recent advances in the molecular dissection of amygdala circuits, we see this as the next important step to understanding interneuron function in the BLA, but beyond the scope of the present work.

However, we have extended our analysis to investigate additional functional properties, which we believe strengthen and support our claims of functional specialisations within heterogeneous interneuron populations. As outlined above (our reply to Comment 2), we have now explored whether SST and VIP interneurons exhibit specific activity reflecting internal fear states. This analysis reveals an important distinction, namely that VIP interneurons are mostly stronger activated by external stimuli (novelty and fear expression, but also aversive events)

but not modulated by internal fear states, suggesting that these interneurons are preferentially mediating cue-driven disinhibition to salient stimuli. In contrast, a high fraction of SST interneurons is additionally modulated during immobility behaviour, and thus encode internal behavioural states, but also contribute to extinction-related inhibition – indicating that these cells could preferentially signal safety conditions.

However, we would like to thank the reviewer for the suggestion on the statistical analysis to support our claim that responses of SST and VIP neurons change over the course of fear learning and extinction. We have considered the reviewers point and updated our analysis in (now) Figure 9c (Rebuttal Figure 3.4) by comparing different timepoints within molecular interneuron subtypes, showing that from 'Before conditioning' to 'After extinction', SST neurons overall increase their CS+ responses, while VIP interneuron CS+ responses decrease.

Rebuttal Figure 3.4: CS+ development in molecular interneuron subtypes. Calcium response as area under the curve (AUC) during CS+ presentations across learning for SST and VIP interneurons SST, n = 114; VIP, n = 152). Paired Wilcoxon test with Bonferroni correction; SST 'Before conditioning' vs. 'After extinction', $p = 0.036$, VIP 'Before conditioning' vs. 'After extinction', $p = 0.012$.

We have further added ideas how this is computed within the local network of distinct inhibitory and excitatory neurons in the discussion. We suggest that the canonical VIP→SST→PN disinhibitory motif in the BLA operates in a dynamic, state-dependent manner, shaped by distinct long-range inputs. VIP interneurons are likely driven by sensory inputs during high-fear states, promoting fear expression through disinhibition of PN dendrites via SST suppression. In contrast, SST interneuron activity in low-fear states may be supported by hippocampal and prefrontal afferents, contributing to fear suppression by promoting PN dendritic inhibition. While external inputs may play a primary role in modulating these populations, local reciprocal inhibition between VIP and SST interneurons likely helps to stabilise their opposing activity patterns across fear states. In parallel, competition between functionally distinct ensembles of excitatory neurons supported by local inhibitory populations – such as PN populations encoding fear and extinction (Herry et al., 2008; Senn et al., 2014; Vogel et al., 2016) – may of course further bias the network's output towards either promoting or suppressing fear, thereby reinforcing state-dependent microcircuit dynamics.

5. There was limited interpretation or speculation about the functional significance of different response types. This creates the impression that this information, at least at this stage, is not necessarily informative about memory-related processes and it is more of an open-ended question.

We have now significantly expanded our analysis and discussion – also based on the additional analyses performed during the revision – to address this point. For example, we find that the size of specific interneuron plasticity cluster fractions correlates with behavioural outcomes (new Supplementary Figure S7a-d, see Rebuttal Figure 3.5). Mice with a greater

number of 'Fear inhibited' CS+ interneurons demonstrated a greater ability to distinguish between predictive CS+ and control CS- cues. This suggests that these cells facilitate the accurate recognition of threats by facilitating transient disinhibition during the expression of fear. Conversely, a greater number of 'Extinction activated' interneurons was linked to poorer discrimination, which may reflect individual differences in fear generalisation, possibly due to a pre-existing inhibitory bias. Furthermore, these two CS+ clusters showed strong modulation of activity during immobility behaviour outside of any auditory cue presentation, which is consistent with an involvement in state transitions. This suggests that these cell types may encode both stimulus-bound associations and more generalised internal fear or safety states, and supports the idea that the BLA contains inhibitory ensembles that bridge sensory-driven and state-dependent aspects of fear regulation. We also added a further interpretation of differences between PN and interneuron plasticity types to the discussion, highlighting the more dynamic, state-dependent coding of interneurons in fear and extinction compared to persistent encoding of fear memory in PNs (Gründemann et al., 2019; Herry et al., 2008; Repa et al., 2001). Our data suggests that interneurons regulate the moment-to-moment expression of fear or safety rather than storing long-term associations. This functional distinction may point to a fundamental organisational principle of fear circuits: PNs act as stable encoders of learned associations, while interneurons provide flexible inhibitory gating of PN output and plasticity. This can enable adaptive switching between fear and safety states, likely through the integration of contextual and top-down inputs, indicating a functional specialisation within the amygdala that supports both stable memory encoding and flexible behavioural expression.

Rebuttal Figure 3.5: Relationship between interneuron plasticity and behavioural performance. a-b, Correlation of fractions of neurons in across-day CS+ plasticity clusters with learning-related performance in individual animals (N = 9) for a, 'Fear inhibited' interneurons and b, 'Extinction activated' interneurons. Lines indicate linear regression fit (shown for visualisation); correlation coefficient ρ reflects Spearman's rank-order correlation.

Reviewer #4 (Remarks to the Author):

This manuscript by Favila et al. investigates the role of interneurons in the amygdala during fear learning and extinction. The study employs a miniscope to track neuronal activity and reports mixed selectivity to different stimuli alongside complex plastic responses. The findings contribute to our understanding of how specific interneuron populations, namely VIP and SST, differentially respond to fear-related cues. In this paper, they conclude that VIP interneurons are predominantly activated in high-fear states, whereas SST interneurons show a preference for safety cues and suppress excitatory neuron responsiveness.

The authors analyze a large population of interneurons over several days, allowing for the study of neuronal dynamics beyond previous work. Interneuron responses appear highly dynamic, showing chance-level mixed selectivity, but after learning, their preference for CS+

over CS- remains stable. VIP interneurons demonstrate stronger activation to the US, with 50% of the population classified as “activated stable CS” cells, and their CS+ activation is subsequently suppressed after extinction. SST interneurons, in contrast, are more inhibited by the US, uniquely display an “activated up CS” category, and show stronger CS+ inhibition, which shifts to activation after extinction. The study suggests that VIP interneurons signal novel and fear-conditioned states, whereas SST interneurons signal safety during extinction.

The study presents novel findings with a robust methodology and well-executed controls, making it compelling and suitable for this journal; however, I have concerns regarding the terminology used and certain conclusions drawn from the data.

1. The term “mixed selectivity” should be clarified, as the study does not demonstrate that these neurons are more mixed-selective than a random population. In particular, since this effect is equal to chance, why do the authors show it as a main result?

We thank the reviewer for raising this point. Indeed, “mixed selectivity” may not be the most appropriate term, as this typically refers to neurons whose activity reflects the conjunction of two or more independent variables, often in a non-linear fashion. We have changed the wording throughout the manuscript to “overlapping encoding of sensory stimuli”, which is more accurate and closer to what we wanted to express.

While the combined neuronal activation for auditory cues and foot shock was at chance level overall, we believe that this result is still relevant: It demonstrates that there is no immediate integration of auditory sensory input and aversive foot shock signals within the BLA interneuron population at the level tested, including SST and VIP interneurons (see new Supplementary Figure S10e-h). This suggests that sensory and aversive information remain largely segregated during this phase of fear conditioning at the level of inhibitory interneurons. We believe this finding is significant because it limits models of how fear-related sensory and aversive signals are processed and integrated in the brain. Rather than assuming the immediate non-linear integration of sensory and shock information, our data suggest that such integration in interneurons may require additional network effects during consolidation. Therefore, reporting this “null result” provides important insight into the neural coding structure during fear conditioning and could inform the interpretation of subsequent studies and analyses that may reveal emergent conjunctive encoding at later stages or under different conditions. We have added these arguments to the manuscript.

2. The statement “CS+ evolved independently of US activation” is somewhat misleading, as CS+ dynamics are dependent on US activation at the population level (even if single CS+ cells can evolve without responding to the US). This distinction should be made clearer.

We have made this clearer in the respective sections (results and discussion) and clearly refer to individual neurons now.

3. While percentages are reported for neuronal responses before and after conditioning, presenting raw numbers would be important to assess whether the sample size remains consistent and to understand the evolution of activation patterns. For example, the statement that “a larger fraction was selectively activated by CS- only before conditioning” could be misleading. The authors should show in parallel the raw numbers of activated neurons for CS+ and CS- before conditioning and compare them to show that these numbers are equal. If the baseline activation of CS- was higher than CS+, this could suggest an intrinsic difference between the tones rather than an experimental effect.

We have added the suggested analysis of neuronal responses before conditioning to Supplementary Figure S5 (new panels g and h; see also Rebuttal Figure 4.1a-c). Here, we are reporting proportions of CS+ and CS- modulated neurons for individual mice, and neither the fractions of activated nor inhibited neurons are different. Notably, the proportion of CS- activated neurons in habituation does not predict the fraction of neurons in the 'CS- activated before conditioning' cluster, indicating that these neurons can be allocated to different clusters upon learning (Rebuttal Figure 4.1d).

Rebuttal Figure 4.1: CS- responses before conditioning. **a**, Proportion of cells in CS+ and CS- clusters (CS+, n = 165; CS-, n = 185). **b**, Proportion of CS+ and CS- activated neurons during habituation (N = 9 mice). **c**, Proportion of CS+ and CS- inhibited neurons during habituation (N = 9 mice). **d**, Proportion of CS- activated cells does not correlate with the size of the 'CS- activated before conditioning' cluster (N = 9 mice). Line indicates linear regression fit (shown for visualisation); correlation coefficient ρ reflects Spearman's rank-order correlation.

To avoid the misleading statement the reviewer pointed out above, we have now changed the sentence in the manuscript to: *“While the predictive CS+ induced significantly stronger activation at the single cell level after conditioning (CS+ 18%, CS- 8%) and after extinction (CS+ 17%, CS- 9%), for the CS-, a larger fraction of interneurons was found in the ‘Activated before conditioning’ cluster (CS+ 18%, CS- 27%; Fig. 5h), an effect that could not simply be attributed to increased neuronal activation by the CS- in naïve mice before learning (Fig. S5f-g).”* (lines 214ff).

Given the differences in recorded cell numbers between animals (37-93 cells per mouse, Supplementary Figure S2b), we are refraining from reporting raw cell numbers when comparing effects across mice. But we have now added raw cell numbers to the Source Data table where appropriate. Furthermore, we would like to clarify that the sample size for individual interneuron types remains constant throughout the comparison of activity patterns across days, since only neurons tracked across all days are included in our analysis.

4. Figures 3 and 4: Could some of the observed differences in interneuron responses be driven by variability between animals rather than within animals? Reporting the percentage of neurons in each cluster per animal might help clarify this.

To illustrate variability between animals, we previously included box-and-whisker plots with individual animal values for clustering results for each stimulus (see for example Figure 3h), and reported the requested values for each animal ID in the Source Data file. We are now additionally providing a new Supplementary Figure S4, which visualises the distribution of US, CS+ and CS- clusters for each animal. The direct comparison of cluster distribution as measured by the neuronal population (n = 519 cells) vs. averaged across animals (N = 9) shows that these are almost identical (Rebuttal Figure 4.2), and further demonstrates that differences in cluster distributions are not driven by individual mice with high cell numbers.

Rebuttal Figure 4.2: Variability of interneuron response types. **a**, Comparison of "across population" analysis for US responsive interneurons in *GAD2-Cre* mice ($n = 393$ cells, with results averaged across animals ($N = 9$)). **b**, Comparison of "across population" analysis for interneurons in CS+ and CS- clusters during fear conditioning (CS+, $n = 297$ cells; CS-, $n = 312$) with results averaged across animals ($N = 9$)). **c**, Comparison of "across population" analysis for interneurons in across-day plasticity clusters (CS+, $n = 365$ cells; CS-, $n = 357$) with results averaged across animals ($N = 9$)).

5. *Figure 5: Would it be beneficial to correlate calcium responses of "high fear state" and "low fear state" neurons within individual mice or across different time points in extinction sessions? Why not also look at the response of neurons over extinction learning as well? This could strengthen the claim that these neurons encode fear states rather than other factors.*

We thank the reviewer for this suggestion. We have now added two additional analyses to address how responses of individual neurons change across extinction learning. Firstly, we are showing how CS+ responses of across-days clusters (fear and extinction neurons) develop over the course of extinction learning. New Figure 5j (Rebuttal Figure 4.3a, for all BLA interneurons) and new Supplementary Figure S13a-b (for SST and VIP interneurons) illustrate that responses of both 'Fear' and 'Fear inhibited' interneuron clusters gradually weaken during the first extinction session but return at a lower level 24 hours later at the beginning of the second session, reflecting spontaneous recovery at the behavioural level. In contrast, extinction neuron responses, both activated and inhibited, developed more rapidly at the end of the second extinction session.

In a complementary approach, we use K-means clustering of all interneurons tracked across days ($n = 519$ GAD2, $n = 114$ SST, $n = 152$ VIP) based on their responses across the two-day extinction training. This revealed similar functional clusters as found when focussing on neurons significantly modulated across habituation, test and extinction – namely fear activated, fear inhibited and extinction activated cells (new Supplementary Figure S6g-h, Rebuttal Figure 4.3b-c). We also detected additional functional subgroups, including a separate group of neurons which were activated predominantly in extinction session 1, while extinction inhibited cells were grouped in a cluster inhibited across all CS+ presentations in extinction session 2. Of note, 'Extinction 1 activated' neuron activity was selectively enriched in VIP interneurons (Rebuttal Figure 4.3d, see also new Supplementary Figure S13c-e), stressing their importance for signalling fear-related external cues. These new results highlight a dynamic, experience-dependent reorganisation of interneuron activity during fear extinction learning.

Rebuttal Figure 4.3: Plasticity of amygdala interneurons in extinction learning. **a**, Development of interneuron responses across extinction sessions in *GAD2-Cre* mice. Averages for four CS+ presentations are shown as indicated. All cells, $n = 519$; Fear, $n = 66$; Fear inhibited, $n = 48$; Extinction, $n = 63$; Extinction inhibited, $n = 33$. **b**, Heatmap of CS+ responses in BLA interneurons from in *GAD2-Cre* mice clustered into groups depending on their response pattern across the two extinction sessions ($n = 519$; Extinction 1 activated, $n = 93$; Fear activated, $n = 116$; Fear inhibited, $n = 109$; Extinction activated, $n = 81$; Extinction inhibited, $n = 120$). **c**, Corresponding proportion of interneurons in CS+ extinction clusters ($n = 519$). **d**, Proportion of cells in CS+ extinction clusters for SST and VIP interneurons (SST, $n = 114$; VIP, $n = 152$) based on recordings from *SST-Cre* and *VIP-Cre* mice. Chi-Square test ($\chi^2(4) = 19.773$), $p = 0.0006$; SST vs. VIP, *post hoc* Chi-Square test with Bonferroni correction, Extinction 1 activated, $p = 0.0028$. Average traces **a** are mean with s.e.m.; ** $p < 0.01$, *** $p < 0.001$.

Minor points:

Line 124: The figure reference appears incorrect; should it be S1b?

The figure reference refers to differences in CS/US responsive cells shown in Fig. 2c and e, where we colour-code individual datapoints for mice with LA and BA implants. For clarity, we have now added the suggested reference to the implant sites as well (now Figure S2b, see line 134 in the manuscript).

Figure 2H: Would adding a correlation test help quantify whether US responses differ significantly from CS+ and CS-?

The reviewer is correct that we have not considered comparing spatial clustering of CS+, CS- and US responsive interneurons. We have therefore changed the illustration in Figure 2g-h (Rebuttal Figure 4.4), showing different examples of spatial maps for responsive neurons separated for CS+, CS- and US responses. Spatial distances between responsive cells are now plotted as cumulative distribution, and a Kolmogorov-Smirnov test is used to show that there is no difference between CS+, CS- and US responsive cells.

Rebuttal Figure 4.4: Spatial mapping of CS+, CS- and US coding neurons. **a**, Example spatial maps of CS+, CS- and US coding neurons. **b**, Cumulative probability distribution of the pairwise distance of CS+, CS- and US modulated neurons, showing no spatial clustering.

Reviewer #5 (Remarks to the Author):

Rebuttal references

- Adler, A., Zhao, R., Shin, M.E., Yasuda, R., and Gan, W.-B. (2019). Somatostatin-Expressing Interneurons Enable and Maintain Learning-Dependent Sequential Activation of Pyramidal Neurons. *Neuron* 102, 202–216.e7.
- d'Aquin, S., Szonyi, A., Mahn, M., Krabbe, S., Gründemann, J., and Lüthi, A. (2022). Compartmentalized dendritic plasticity during associative learning. *Science* 376, eabf7052.
- Artinian, J., and Lacaille, J.-C. (2018). Disinhibition in learning and memory circuits: New vistas for somatostatin interneurons and long-term synaptic plasticity. *Brain Res. Bull.* 141, 20–26.
- Bastos, G., Holmes, J.T., Ross, J.M., Rader, A.M., Gallimore, C.G., Wargo, J.A., Peterka, D.S., and Hamm, J.P. (2023). Top-down input modulates visual context processing through an interneuron-specific circuit. *Cell Rep.* 42, 113133.
- Chevy, Q., Szadai, Z., Hertäg, L., Moll, M., Gibson, E.T., Costa, R.P., and Kepecs, A. (2024). A Cortical Microcircuit for Region-Specific Credit Assignment in Reinforcement Learning. *BioRxiv*.
- Cichon, J., and Gan, W.-B. (2015). Branch-specific dendritic Ca(2+) spikes cause persistent synaptic plasticity. *Nature* 520, 180–185.
- Gentet, L.J., Kremer, Y., Taniguchi, H., Huang, Z.J., Staiger, J.F., and Petersen, C.C.H. (2012). Unique functional properties of somatostatin-expressing GABAergic neurons in mouse barrel cortex. *Nat. Neurosci.* 15, 607–612.
- Gründemann, J., Bitterman, Y., Lu, T., Krabbe, S., Grewe, B.F., Schnitzer, M.J., and Lüthi, A. (2019). Amygdala ensembles encode behavioral states. *Science* 364.
- Hájos, N. (2021). Interneuron types and their circuits in the basolateral amygdala. *Front. Neural Circuits* 15, 687257.
- Herry, C., Ciocchi, S., Senn, V., Demmou, L., Müller, C., and Lüthi, A. (2008). Switching on and off fear by distinct neuronal circuits. *Nature* 454, 600–606.
- Hochgerner, H., Singh, S., Tibi, M., Lin, Z., Skarbianskis, N., Admati, I., Ophir, O., Reinhardt, N., Netser, S., Wagner, S., et al. (2023). Neuronal types in the mouse amygdala and their transcriptional response to fear conditioning. *Nat. Neurosci.* 26, 2237–2249.
- Karnani, M.M., Jackson, J., Ayzenshtat, I., Tucciarone, J., Manoocheri, K., Snider, W.G., and Yuste, R. (2016b). Cooperative subnetworks of molecularly similar interneurons in mouse neocortex. *Neuron* 90, 86–100.
- Karnani, M.M., Jackson, J., Ayzenshtat, I., Hamzehei Sichani, A., Manoocheri, K., Kim, S., and Yuste, R. (2016a). Opening holes in the blanket of inhibition: localized lateral disinhibition by VIP interneurons. *J. Neurosci.* 36, 3471–3480.
- Kepecs, A., and Fishell, G. (2014). Interneuron cell types are fit to function. *Nature* 505, 318–326.
- Kim, J., Pignatelli, M., Xu, S., Itohara, S., and Tonegawa, S. (2016). Antagonistic negative and positive neurons of the basolateral amygdala. *Nat. Neurosci.* 19, 1636–1646.
- Krabbe, S., Paradiso, E., d'Aquin, S., Bitterman, Y., Courtin, J., Xu, C., Yonehara, K., Markovic, M., Müller, C., Eichlisberger, T., et al. (2019). Adaptive disinhibitory gating by VIP interneurons permits associative learning. *Nat. Neurosci.* 22, 1834–1843.
- Kullander, K., and Topolnik, L. (2021). Cortical disinhibitory circuits: cell types, connectivity and function. *Trends Neurosci.* 44, 643–657.
- Manoocheri, K., and Carter, A.G. (2022). Rostral and caudal basolateral amygdala engage distinct circuits in the prelimbic and infralimbic prefrontal cortex. *Elife* 11.
- Perumal, M.B., and Sah, P. (2021). Inhibitory circuits in the basolateral amygdala in aversive learning and memory. *Front. Neural Circuits* 15, 633235.
- Pi, H.-J., Hangya, B., Kvitsiani, D., Sanders, J.I., Huang, Z.J., and Kepecs, A. (2013). Cortical interneurons that specialize in disinhibitory control. *Nature* 503, 521–524.
- Piet, A., Ponvert, N., Ollerenshaw, D., Garrett, M., Groblewski, P.A., Olsen, S., Koch, C., and Arkhipov, A. (2024). Behavioral strategy shapes activation of the Vip-Sst disinhibitory circuit in visual cortex. *Neuron* 112, 1876–1890.e4.
- Reeb, Z., Magyar, D., Weisz, F., Fekete, Z., Muller, K., Vikor, A., Peterfi, Z., Andrasi, T., Veres, J., and Hajos, N. (2023). Diversity and connectivity of principal neurons in the lateral and basal nuclei of the mouse amygdala. *BioRxiv*.
- Repa, J.C., Muller, J., Apergis, J., Desrochers, T.M., Zhou, Y., and LeDoux, J.E. (2001). Two different lateral amygdala cell populations contribute to the initiation and storage of memory. *Nat. Neurosci.* 4, 724–731.
- Senn, V., Wolff, S.B.E., Herry, C., Grenier, F., Ehrlich, I., Gründemann, J., Fadok, J.P., Müller, C., Letzkus, J.J., and Lüthi, A. (2014). Long-range connectivity defines behavioral specificity of amygdala neurons. *Neuron* 81, 428–437.
- Vogel, E., Krabbe, S., Gründemann, J., Wamsteeker Cusulin, J.I., and Lüthi, A. (2016). Projection-Specific Dynamic Regulation of Inhibition in Amygdala Micro-Circuits. *Neuron* 91, 644–651.

Wolff, S.B.E., Gründemann, J., Tovote, P., Krabbe, S., Jacobson, G.A., Müller, C., Herry, C., Ehrlich, I., Friedrich, R.W., Letzkus, J.J., et al. (2014). Amygdala interneuron subtypes control fear learning through disinhibition. *Nature* 509, 453–458.

Zhang, J., Hughes, R.N., Kim, N., Fallon, I.P., Bakhurin, K., Kim, J., Severino, F.P.U., and Yin, H.H. (2023). A one-photon endoscope for simultaneous patterned optogenetic stimulation and calcium imaging in freely behaving mice. *Nat. Biomed. Eng.* 7, 499–510.

Zhang, X., Kim, J., and Tonegawa, S. (2020). Amygdala reward neurons form and store fear extinction memory. *Neuron* 105, 1077–1093.e7.

Rebuttal Letter

Heterogeneous plasticity of amygdala interneurons in associative learning and extinction

Structure: Reviewer comments: *black*; our replies: *blue*

REVIEWERS' COMMENTS

Reviewer #1 (Remarks to the Author):

I thank the authors for their thoughtful revisions. The new data and analyses significantly enhance the study's impact and conceptual depth, particularly the single-cell analyses linking neural activity to behavior.

While I am satisfied with the revisions, I maintain that the core finding on heterogeneous BLA interneuron plasticity, though well-executed, builds directly on concepts from prior acute and population-level studies.

Furthermore, the authors have not addressed the major concern raised by Reviewer #3 regarding the novel finding mentioned at the end of the first paragraph.

We would like to thank the reviewer for the positive assessment of our revised manuscript.

The reviewer is correct that the present study builds on our previous work on interneuron activity in aversive learning. Yet, as pointed out before by us in the previous round of revision, none of the publications cited by the reviewer (our publications Wolff et al., 2014; Krabbe et al., 2019) has addressed across-day plasticity of individual amygdala interneurons, and no additional citations are provided by the reviewer here that would support the statement of the reviewer. Our present work is novel and the first in-depth study showing that individual amygdala interneurons exhibit heterogeneous and complex plastic responses across days during the acquisition, expression and extinction of aversive memories. We further show that distinct interneuron subtypes (VIP and SST) differentially encode external sensory cues and internal fear states, revealing mechanisms of disinhibition and state-dependent inhibition that have not been previously described. These results establish inhibitory interneurons as active, plastic components of the amygdala memory circuit, fundamentally extending current models of how emotional memories are encoded and regulated. This also relates to the remark by Reviewer 1 that a concern of Reviewer 3 regarding “novelty” was not fully addressed. Here, we would like to further note that Reviewer 3 explicitly confirmed in their follow-up comments that we fully addressed any of their concerns in our revisions.

Reviewer #2 (Remarks to the Author):

Reviewer #3 (Remarks to the Author):

The authors have addressed my concerns and the new data provided help strengthen the impact.

We would like to thank the reviewer for the positive evaluation of our work and for the comments along the way which helped improving our manuscript.

Reviewer #4 (Remarks to the Author):

The authors have satisfactorily addressed all previous concerns, and the manuscript has improved significantly. I recommend publication in its current form.

We would like to thank the reviewer for the positive evaluation of our work and for the comments along the way which helped improving our manuscript.

Reviewer #5 (Remarks to the Author):
